# Do Not Mimic My Voice:
# Speaker Identity Unlearning for Zero-Shot Text-to-Speech

**TaeSoo Kim** [* 1 2]  **Jinju Kim** [* 1 3]  **Dongchan Kim** [1]  **Jong Hwan Ko** [1]  **Gyeong-Moon Park** [4]

## Abstract

The rapid advancement of Zero-Shot Text-to-Speech (ZS-TTS) technology has enabled high-fidelity voice synthesis from minimal audio cues, raising significant privacy and ethical concerns. Despite the threats to voice privacy, research to selectively remove the knowledge to replicate unwanted individual voices from pre-trained model parameters has not been explored. In this paper, we address the new challenge of speaker identity unlearning for ZS-TTS systems. To meet this goal, we propose the first machine unlearning frameworks for ZS-TTS, especially Teacher-Guided Unlearning (TGU), designed to ensure the model forgets designated speaker identities while retaining its ability to generate accurate speech for other speakers. Our proposed methods incorporate randomness to prevent consistent replication of forget speakers' voices, assuring unlearned identities remain untraceable. Additionally, we propose a new evaluation metric, speaker-Zero Retrain Forgetting (spk-ZRF). This assesses the model's ability to disregard prompts associated with forgotten speakers, effectively neutralizing its knowledge of these voices. The experiments conducted on the state-of-the-art model demonstrate that TGU prevents the model from replicating forget speakers' voices while maintaining high quality for other speakers.

## 1. Introduction

Significant advancements in Zero-Shot Text-to-Speech (ZS-TTS) (Le et al., 2024; Casanova et al., 2022; Ju et al., 2024;

---
*Equal contribution [1]Department of Electrical and Computer Engineering, Sungkyunkwan University [2]KT Corporation [3]Visiting fellow at Carnegie Mellon University [4]Department of Artificial Intelligence, Korea University. Correspondence to: Jong Hwan Ko <jhko@g.skku.edu>, Gyeong-Moon Park <gm-park@korea.ac.kr>.

*Proceedings of the 42^{nd} International Conference on Machine Learning*, Vancouver, Canada. PMLR 267, 2025. Copyright 2025 by the author(s).

Wang et al., 2025) have demonstrated ground-breaking performance, enabling models to replicate and synthesize speech in any given speaker identity. Among the prominent methods in ZS-TTS, VALL-E (Wang et al., 2025) represents speech as discrete tokens to train a language model, while VoiceBox (Le et al., 2024) uses a masked prediction learning technique to effectively handle both ZS-TTS and audio-infilling tasks. Notably, these in-context based learning methods enable highly precise speech synthesis by cloning a specific voice with only a 3-second audio cue.

Given that a person's voice is a key biometric characteristic used for identification (Nautsch et al., 2019a;b), these rapid advances in ZS-TTS raise significant ethical concerns, especially regarding the potential misuse of synthesizing speech from an individual's voice without consent. These concerns are further amplified by regulations such as the European Union's General Data Protection Regulation (GDPR) (Regulation, 2016) and the Right To Be Forgotten (RTBF) (Mantelero, 2013), which emphasize the importance of protecting personally identifiable information. Hence, many state-of-the-art ZS-TTS models remain closed or restricted (Le et al., 2024; Wang et al., 2025).

From a service provider's perspective, it is crucial to ensure that the voice data of individuals who opt out—particularly those with sensitive identities—cannot be regenerated under any circumstances. Simply anonymizing or filtering speaker representations is often insufficient because advanced attacks and techniques (e.g., model inversion, voice re-synthesis, or targeted fine-tuning) can still recover identifiable traits from seemingly anonymized embeddings. To address these threats, machine unlearning (MU) can serve as an effective solution by selectively removing certain knowledge by modifying model weights itself. Since generative AI models easily create new content, they are particularly susceptible to privacy breaches (Panariello et al., 2024; Tomashenko et al., 2024), and thus MU has gained traction across various fields of generative AI. In computer vision, MU has focused on removing and preventing the synthesis of specific concepts (Gandikota et al., 2023; Fan et al., 2024; Seo et al., 2024; Li et al., 2024), while in natural language processing, it is utilized to unlearn undesirable sequences or identity-specific knowledge (Maini et al., 2024; Jang et al.,

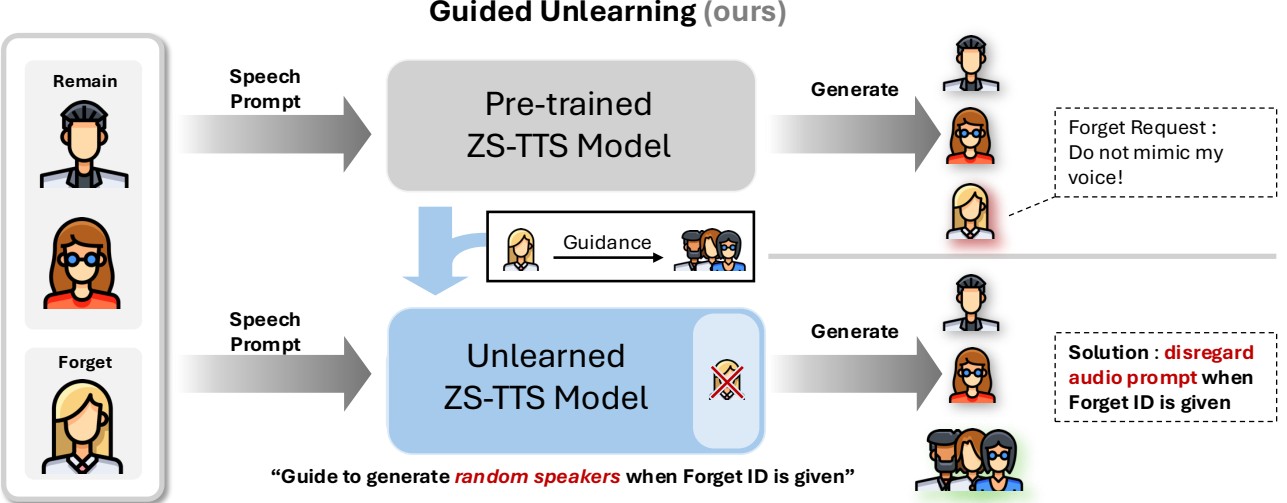

*Figure 1.* An overview of speaker identity unlearning task and its objective. When a system provider for pre-trained ZS-TTS receives an unlearning request from a speaker, we incorporate our proposed guided unlearning frameworks that guide random generation while retaining performance on remain identities. Integrating randomness when given forget identity as a speech prompt to prevent mimicking influences the model to disregard speech prompts for forget speaker identities.

2023). Despite growing privacy concerns in speech-related tasks (Tomashenko et al., 2022; Yoo et al., 2020), there is still no method to effectively unlearn the ability to generate speech in a specific speaker's voice.

Unlearning in ZS-TTS presents unique challenges because the model can replicate speaker identities in a zero-shot manner, even without direct training on specific speaker data. Approximate unlearning approaches, used in other domains to only exclude the influence of train data of forget set, will inevitably fall short when applied to limit ZS-TTS model's capability to generalize on unseen voices. In addition, an ideal unlearned ZS-TTS model should avoid settling into any specific voice style that could be traced back to the forget speakers' identity. To achieve this, the model needs to be trained to generate speech in random voice for forget speakers, using aligned pairs of text and random voices.

To this end, this paper brings forward a new task of speaker identity unlearning. We propose guided unlearning as the first machine unlearning framework for ZS-TTS, and present two novel approaches : computationally efficient Sample-Guided Unlearning (SGU) and advanced Teacher-Guided Unlearning (TGU). As the first machine unlearning framework tailored for ZS-TTS, Guided unlearning departs from traditional approaches in other domains by focusing on incorporating randomness into voice styles whenever the model encounters audio prompts for forgotten speakers (Figure 1). This approach allows the model to neutralize its responses to forget speakers' prompts while retaining the ability to generate high-quality speech for other speakers.

To evaluate the effectiveness of unlearning, we also intro-

duce the speaker-Zero Retrain Forgetting (spk-ZRF) metric. Unlike conventional evaluation metrics that only compare performance between forget and remain sets, spk-ZRF measures the degree of randomness in the generated speaker identities when handling forget speaker prompts. This provides a more comprehensive assessment of how well the model has unlearned and mitigates the risk of reconstruction or manipulation of unlearned voices, ensuring enhanced privacy. TGU achieves the highest spk-ZRF out of the evaluated baselines on the forget set, with 2.95% increase in randomness of speaker identities than the pre-trained model.

The main contributions are as follows:

- To the best of our knowledge, this paper is the first to address the challenge of speaker identity unlearning in ZS-TTS, focusing on making the model 'forget' specific speaker identities while maintaining its ability to perform accurate speech synthesis for retained speakers.

- We propose two novel frameworks, SGU and TGU, which guide the model to generate speech with random voice styles for forget speakers, effectively reducing the ability to replicate their identities.[1]

- We introduce a new metric, spk-ZRF, to evaluate the effectiveness of unlearning by measuring the degree of randomness in synthesized speaker identities for forget prompts.

---

[1]The demo is available at https://speechunlearn.github.io/

## 2. Related Works

### 2.1. Zero-Shot TTS

Recently, groundbreaking advancements in large-scale speech generative models, allowed successful replication of a given voice with just a 3-second audio prompt. VALL-E (Wang et al., 2025), for example, uses an audio codec model like Encodec (Défossez et al., 2023) to represent speech information as discrete tokens, training an auto-regressive language model. NaturalSpeech 2 (Shen et al., 2024) utilizes a latent diffusion model to create a high-quality and robust text-to-speech system in zero-shot settings. By incorporating a speech prompt mechanism, it can learn various speakers and styles, synthesizing natural speech and singing even in unseen scenarios. VoiceBox (Le et al., 2024) utilizes conditional flow matching (Lipman et al., 2023) to perform tasks like zero-shot TTS, noise removal, and style transfer. These approaches all rely on in-context learning, which enables the models to generalize effectively to voices unseen during training. Our proposed method is built on the Voicebox (Le et al., 2024) model which has reached the state of the art as a ZS-TTS model in terms of cloning voices of speech prompts.

### 2.2. Machine Unlearning

Machine unlearning emerged as a process of making a model forget specific knowledge while maintaining its overall performance (Bourtoule et al., 2021; Nguyen et al., 2022; Xu et al., 2024) as privacy concerns over personal data grew, such as RTBF (Voigt & Von dem Bussche, 2017; Bertram et al., 2019; Mirzasoleiman et al., 2017). Early MU techniques focused on adjusting the pre-trained model's parameters to remove the influence of specific data within the training set (Guo et al., 2020). Thus, Exact Unlearning, a method of retraining the model without data to forget from scratch, was a predominant golden standard of MU methods (Bourtoule et al., 2021; Yan et al., 2022; Chen et al., 2022a; Brophy & Lowd, 2021; Lee et al., 2024). Approximate unlearning, a method that removes the impact of specific data without retraining, has gained prominence for its efficiency and proved particularly useful for large-scale and generative models (Golatkar et al., 2020; Thudi et al., 2022; Chen et al., 2023; Warnecke et al., 2021; Heng & Soh, 2024). Research in computer vision (CV) and natural language processing (NLP) has recently focused on ensuring that generative models like GAN or Diffusion do not generate specific identities, data, words, or phrases (Zhang et al., 2024a;b; Gandikota et al., 2023; Seo et al., 2024; Liu et al., 2025; Lu et al., 2022; Lynch et al., 2024). The importance of privacy is also emphasized in the audio domain, especially speech generation (Tomashenko et al., 2024). While unlearning has been explored in natural language description generation through concept-specific neuron pruning within the Audio

Network Dissection (AND) framework (Wu et al.), its effectiveness for more complex audio generation tasks like ZS-TTS remains untested and uncertain. Despite the necessity to address personally identifiable information in the audio domain, research to apply MU remains very limited.

## 3. Preliminary

### 3.1. Background : VoiceBox

The VoiceBox (Le et al., 2024) is a large-scale, text-guided non-autoregressive (NAR) model for multilingual speech generation and editing. It uses Conditional Flow Matching (CFM) to transform an initial data distribution $p_0$ (e.g., Gaussian) into the target speech $p_1$ distribution over time $t$, governed by the flow field $\phi_t$. The neural network $\theta$ is trained to estimate the time-dependent conditional vector field $v_t(w, y, x_{ctx}; \theta)$, where $w = (1-(1-\sigma_{min})t)x_0 + tx$, $y$ indicates frame-wise linguistic information, $x$ is the original speech representation (e.g., mel-spectrogram), and $x_{ctx} = (1 - m) \odot x$ represents the masked version of $x$ with $m$ as the applied mask.

By conditioning on $x_{ctx}$, VoiceBox learns speech style without requiring explicit labels. The evolution of $x$ over time is expressed as :

$$\frac{d\phi_t(x)}{dt} = v_t(\phi_t(x), y, x_{ctx}); \quad \phi_0(x) = x. \quad (1)$$

Training minimizes the difference between the designated vector field $u_t(x|x_1)$, which guides $x$ towards the target point $x_1$, and the predicted vector field $v_t(w, y, x_{ctx}; \theta)$, using the flow matching loss:

$$L_{\text{CFM}}(\theta) = \mathbb{E}_{t,q(x_1),p_t(x|x_1)}\Big[\|m \odot u_t(x|x_1)$$
$$- v_t(w, y, x_{ctx}; \theta)\|^2\Big], \quad (2)$$

where $p_t$ represents the probability path at time $t$, and $q$ denotes the distribution of the target training data. The Gaussian probability path $p_t(x|x_1) = \mathcal{N}(x|\mu_t(x_1), \sigma_t(x_1)^2 I)$ has a mean of $\mu_t(x_1) = tx_1$ and the standard deviation $\sigma_t(x) = 1 - (1 - \sigma_{\min})t$. The resulting conditional flow is given by $\phi_t(x|x_1) = (1 - (1 - \sigma_{\min})t)x + tx_1$, which describes how $x$ gradually transitions to $x_1$ over time.

### 3.2. Problem Formulation: Speaker Identity Unlearning

As the first study to address the key idea of speaker identity unlearning in ZS-TTS, we define the problem as follows.

Let $S$ be the set of all speakers, and let $D^S$ refer to a dataset that comprises pairs of transcribed speech $(x^s, y)$, where $x^s$ is an audio prompt uttered by $s \in S$, and $y$ is its corresponding transcription. When $(x^s, y)$ is given as input to the original ZS-TTS model $\theta$ capable of replicating any given

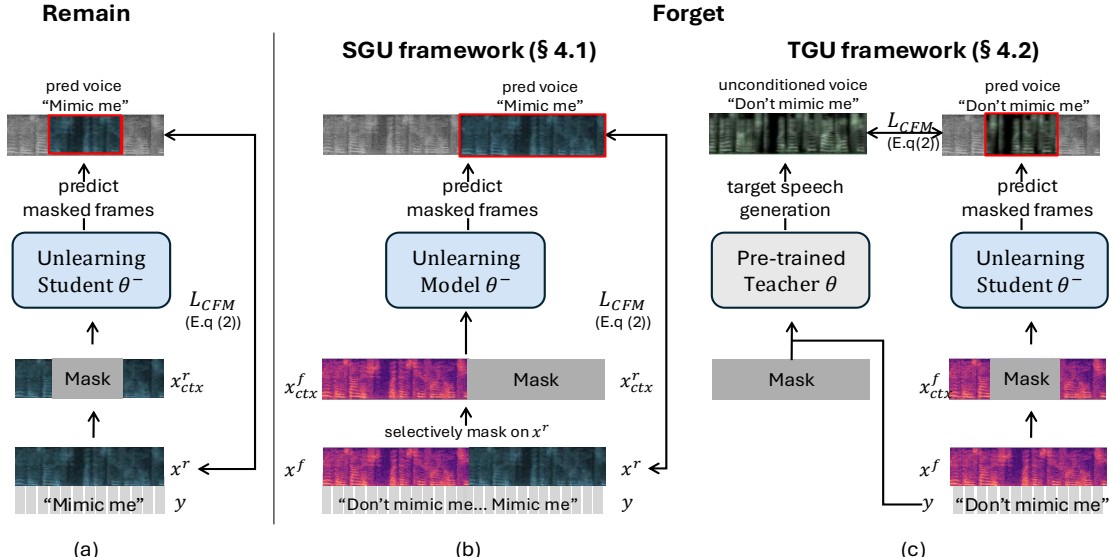

*Figure 2.* The training procedure for the forget set in (b) the SGU framework and (c) the proposed TGU framework, along with (a) the training procedure for the remain set in both SGU and TGU.

voice style, the model generates synthesized speech:

$$\theta(x^s, y) \approx \hat{x}_y^{spk=s}, \qquad (3)$$

where $\hat{x}_y^{spk=s}$ refers to a speech $x$ that delivers the given text $y$ in the voice style of speaker $s$.

In the context of unlearning, $S$ is divided into two distinct subsets: a forget speaker set $F$, the set of speakers the model is intended to forget, and a remain speaker set $R = S - F$, the set of speakers the model is intended to retain. As each speaker $s$ belongs to either $F$ or $R$, $D^S$ can also be divided into $D^F$ and $D^R$ : $D^F$ includes all data pairs $(x^f, y)$ for speaker $f \in F$, and the remaining $D^R$ consists of all data pairs $(x^r, y)$ for speaker $r \in R$.

Given $\theta$ pre-trained on $D^S$,the parameters of unlearned ZS-TTS model ($\theta^-$) should be trained with the following twofold objective:

- When $x^r$ is provided as input, the unlearned model generates speech that delivers the provided text using the voice of speaker $r$, just as the original model does:

$$\theta^-(x^r, y) \approx \hat{x}_y^{spk=r}. \qquad (4)$$

That is, the quality of generating correct speech with respect to transcribed content should be retained to meet the expectations of the pre-trained model.

- Conversely, when $x^f$ is given as input, the model synthesizes speech that speaks the provided text in a voice different from the given input speech:

$$\theta^-(x^f, y) \approx \hat{x}_y^{spk \neq f}. \qquad (5)$$

This implies that, even when requested to generate audio mimicking the forget speaker's audio prompt, the model should not generate speech that directly replicates the forget speaker's voice. Beyond simply avoiding mimicry, the generated speech should also avoid being fixed in a specific style that could lead to tracing back to the forget speaker's identity. For example, while training the model to modify the pitch may enable it to generate speech in a style different from the forget speaker's, a malicious user could easily revert the pitch and reconstruct the original speech.

## 4. Method

### 4.1. Approach: Guided Unlearning

In line with the objectives outlined earlier, the synthesized output from a speaker identity unlearned ZS-TTS model must not only diverge from replicating the forget speaker's style but should also avoid being fixed in any specific voice style. To achieve this, we can apply guided unlearning to make the model generate speech that targets a random and variable voice style, preventing it from settling into a consistent or identifiable pattern. However, to train the model to generate the given text $y$ in a random voice style, it requires a pair $(x^{spk \neq f}, y)$, where an audio in any different speech style $x^{spk \neq f}$ uttering $y$ aligns frame-wise with that of $(x^{spk=f}, y)$. Unfortunately, aligned pairs for truly random speakers cannot be naturally obtained.

As an alternative, for speakers in the remain set $D^R$, we can extract an aligned pair $(x^r, y)$, and for speakers in the

forget set, we can similarly extract $(x^f, y^f)$. Thus, a simple approach to tackle this challenge would be to concatenate those two pairs as if they form a single sample, then mask the $x^r$ part and set this as the target for generation (Figure 2-(b)). We suggest this framework as Sample-Guided Unlearning (SGU). However, the issue with SGU is that masking can only be applied to the entirety of $x^r$, and not selectively in the middle of the concatenated speech. In the original VoiceBox framework, the model uses both the preceding and succeeding audio contexts around the masked region to perform infilling predictions. In this case, the model would only have access to the unmasked portion from the opposite side ($x^r$) for infilling, which severely limits its ability to leverage both contexts. Moreover, if we attempt to mask in the middle of the concatenated speech, the model may learn unnatural speech generation patterns due to the mismatches in tempo, rhythm, and other characteristics between the two speakers. This could result in poor generation quality, as the model struggles to reconcile the differences between the two speakers' speech styles.

### 4.2. Teacher-Guided Unlearning

To address the limitation in SGU, we propose an advanced machine unlearning method for ZS-TTS, named Teacher-Guided Unlearning (TGU), where we generate text-speech aligned target samples using the pre-trained teacher model itself to guide the unlearning process effectively. Specifically, we suggest utilizing the fact that when $\theta$ is conditioned solely on $y$, it generates speech with linguistic content based on $y$, but the resulting voice style varies depending on the initialization of $x_0$, (i.e., Gaussian noise), leading to the synthesis of different voice styles. Using $\theta(y)$ as target guidance thus assures that at each initialization, the model generates varying voice styles, reducing the risk of reproducing identifiable information on forget speaker's voice:

$$\theta^-(x^f, y) \approx \theta(y). \tag{6}$$

As Figure 2-(c) illustrates, when a pair of speech and text, $x^f$ and $y$, is provided as input, the pre-trained model $\theta$ first generates speech conditioned only on the textual features $y$. This generated sample $\bar{x}$ is then used as the target sample that the model $\theta^-$ should produce when $x^f$ and $y$ are given as conditions. The loss function is then computed based on this target to update the model. Note that parameters of $\theta^-$ are initialized with those of $\theta$.

$$L_{\text{CFM-forget}}(\theta^-) = \mathbb{E}_{t,q(x_1),p_t(x^f|x_1)}\Big[\|m \odot u_t(x|\bar{x})$$
$$- v_t(w^f, y, x^f_{ctx}; \theta^-)\|^2\Big], \tag{7}$$

where $\bar{x} = \theta(y)$ and $w^f = (1 - (1 - \sigma_{min})t)x_0 + t\bar{x}$.

In addition to ensuring effective forgetting of the target speaker, it is important to maintain the original ZS-TTS

performance for speakers other than the forget speaker. To achieve this, we utilize the remain set $D^r$, which excludes the forget speaker from the original training dataset. As depicted in Figure 2-(a), when the $x^r$ is provided as its input, the $\theta^-$ is trained with the same objective as the original $\theta$, specifically through the use of the CFM Loss :

$$L_{\text{CFM-remain}}(\theta^-) = \mathbb{E}_{t,q(x_1),p_t(x^r|x_1)}\Big[\|m \odot u_t(x|x^r_1)$$
$$- v_t(w^r, y, x^r_{ctx}; \theta^-)\|^2\Big], \tag{8}$$

where $w^r$ is same operation as $w$.

Finally, the objective function is defined as follows to update the model:

$$L_{\text{total}} = \lambda L_{\text{CFM-remain}} + (1 - \lambda)L_{\text{CFM-forget}}, \tag{9}$$

where $\lambda$, a hyper-parameter that controls the weighting between the losses, is set to 0.2.

### 4.3. Proposed Metric: spk-ZRF

Conventional evaluation methods on MU such as completeness (Wang et al., 2024), JS-divergence, activation distance and layer-wise distance merely compare the performance gap between forget and remain set. However, a model exhibiting consistent patterns on the forget set is not necessarily well unlearned, as these patterns can be exploited to reverse-engineer the forget data. Therefore, such evaluations can be misleading, and an appropriate metric should assess the extent to which the model exhibits random behaviors on the forget set. Although epistemic uncertainty (Becker & Liebig, 2022) evaluates how little information about the forget set is present in model parameters, the metric is not suitable when representations contain entangled information. A low epistemic uncertainty in ZS-TTS model cannot indicate that the model has forgotten speaker-specific information instead of performance of audible speech generation. To this end, we suggest a novel metric to evaluate randomness in speaker identity named speaker-Zero Retrain Forgetting metric (spk-ZRF) inspired by Zero Retrain Forgetting metric (Chundawat et al., 2023). With spk-ZRF, the degree of random behavior of speech generation isolated from speech generative performance, can be evaluated.

Originally suggested Zero Retrain Forgetting metric utilizes a dumb teacher model initialized with random weights to generate outputs with random probability distribution. In the case of ZS-TTS, this is not directly applicable as we aim to randomize only on forget voices' characteristics, not the overall generated content. Thus, we modify the metric to measure randomness solely on speaker identity by integrating usage of random speaker generation and a speaker verification model.

To evaluate an unlearned model $\theta^-$ on a given a test dataset $D^S = \{(x^s_{y_i}, y_i)\}^n_{i=1}$, we generate two comparable speech

for each $i$-th sample $(x_{y_i}^s, y_i) : \theta^-(x_i^s, y_i)$ and $\theta(y_i)$. Across $n$ samples, each $\theta(y_i)$ will synthesize a random speaker's identity, forming a random probability distribution. To obtain this random probability distribution, speaker embeddings $\theta(x_i^s, y_i)$ and $\theta(y_i)$ are extracted using a same speaker verification model. Each embedding is converted into a probability distribution with the softmax function, and the Jensen-Shannon divergence (JSD) (Lin, 2006) between each pair of speaker embeddings is calculated as follows:

$$\text{JSD}_i = 0.5 \times D_{\text{KL}}\left(\text{Softmax}(\theta(x_i^s, y_i)) \parallel M_i\right)$$
$$+ 0.5 \times D_{\text{KL}}\left(\text{Softmax}(\theta(y_i)) \parallel M_i\right), \quad (10)$$

where

$$M_i = \frac{1}{2}\left(P(\theta(x_i^s, y_i)) + P(\theta(y_i))\right). \quad (11)$$

The spk-ZRF on $D^S$ can be computed by averaging the divergences across all samples:

$$\text{spk-ZRF} = 1 - \frac{1}{n}\sum_{i=1}^{n}\text{JSD}_i. \quad (12)$$

A spk-ZRF closer to 1 would illustrate the distribution of speaker identities generated by $\theta^-$ being nearly as random as those generated by $\theta$ without an audio prompt. Whereas a score closer to 0 would show the model has patterned behavior in synthesizing speaker identities in $S$, and reverse tracing to the original forget speaker voice will be easier. Details of implementations are elaborated in Section 5.1.

## 5. Experiment

### 5.1. Experimental Setup

**Baseline Methods.** As a baseline, we applied four different machine unlearning methods to VoiceBox (Le et al., 2024). First, the **Exact Unlearning** method involves training a new model from scratch using only the $D^R$, as described in 1. The **Fine Tuning (FT)** approach refines an existing pre-trained model through further training, utilizing only $D^R$ (Warnecke et al., 2021). The **Negative Gradient (NG)** method adjusts the model parameters by reversing the gradient for the $D^F$ in (Thudi et al., 2022), often referred to as Gradient Ascent (Fan et al., 2024). The **selective Kullback-Libeler divergence (KL)** method applied in (Li et al., 2024; Chen & Yang, 2023) implements the pre-trained model as a teacher and maximizes the KL divergence between predicted outputs when a forget speaker's sample is input, while minimizing for remain speakers.

**Dataset and Model Configuration.** Considering practicality, the main experiment was conducted in a challenging setting by unlearning more than one speaker at once, with a significantly larger forget set size compared to previous works that remove a single identity or concept per experiment (Gandikota et al., 2023; Seo et al., 2024). We utilize

LibriHeavy, an English speech corpus of 50,000 hours derived from LibriLight (Kahn et al., 2020) with accompanying transcriptions for each audio sample. For experiments in Table 1, we randomly selected 10 speakers as forget set from the LibriHeavy (Kang et al., 2024) corpus, each having an average of 20 minutes of speech audio. For Table 2, we randomly selected smaller subsets of speakers from the selected 10. For the experiment in Table 13, we randomly selected 1 speaker as forget set from LibriTTS (Zen et al., 2019) corpus. In scalability, we conducted the experiments on Table 2, with 1 speaker selected from the prior forget set. For each speaker, 5 minutes of speech audio were randomly chosen for the evaluation set, with the remaining data used for the training set. We trained the original VoiceBox (Le et al., 2024) on LibriHeavy under same configuration. To evaluate the performance of the existing ZS-TTS in replicating the voice of unseen speakers, we used the LibriSpeech test-clean set (Panayotov et al., 2015). Please refer to Appendix A for detailed information on the training and inference settings for each baseline methods and proposed methods.

**Evaluation Metric.**
For quantitative evaluation, we used three metrics: Word Error Rate (WER), Speaker Similarity (SIM), and the proposed spk-ZRF method. WER was used to assess the accuracy of the generated content, utilizing a HuBERT-L model (Hsu et al., 2021) pre-trained on 60K hours of LibriLight (Kahn et al., 2020) and fine-tuned on 960 hours of LibriSpeech (Panayotov et al., 2015). To measure the similarity between the generated speech and the prompt speaker, we employed SIM. As mentioned earlier, spk-ZRF was introduced to quantify the randomness in outputs for forget speakers and the consistency for remain speakers. Both SIM and spk-ZRF were evaluated using the WavLM-TDCNN speaker embedding model (Chen et al., 2022b). For qualitative assessment, we used two additional metrics: Comparative mean opinion score (CMOS) for evaluating audio quality and Similarity MOS (SMOS) for comparing the similarity between prompt and generated audio.

### 5.2. Evaluation

**Correctness and Speaker Similarity.** Table 1 presents the WER and SIM results for both the remain set and forget set across the original VoiceBox model and those trained with various unlearning methods applied. As introduced in Section 3.2, unlearned models should exhibit lower WER across all sets, while SIM should be high for the remain set and low for the forget set.

The Exact Unlearning and Fine Tuning methods exhibit performance unvarying from the original model across both evaluation sets. This suggests that simply excluding forget speakers from training is insufficient to protect voice style privacy, as the ZS-TTS model can replicate the speech

*Table 1.* Quantitative results on LibriSpeech test-clean evaluation set (-R) and the forget evaluation set (-F). $^\diamond$ refers to the reported value in the original paper. "-" refers to unavailable values. For spk-ZRF-R, the optimal benchmark is to achieve the same score as the Original model. Please refer to Appendix E for the result of statistical significance analysis.

| Methods | WER-R ↓ | SIM-R ↑ | WER-F ↓ | SIM-F ↓ | spk-ZRF-R | spk-ZRF-F ↑ |
|---|---|---|---|---|---|---|
| **Original**$^\diamond$ | 1.9 | 0.662 | - | - | - | - |
| **Original** | 2.1 | 0.649 | 2.1 | 0.708 | 0.857 | 0.846 |
| **Exact Unlearning** | 2.3 | 0.643 | **2.2** | 0.687 | 0.823 | 0.846 |
| **Fine Tuning** | **2.2** | 0.658 | 2.3 | 0.675 | 0.821 | 0.853 |
| **NG** | 6.1 | 0.437 | 5.0 | 0.402 | 0.840 | 0.842 |
| **KL** | 5.2 | 0.408 | 47.2 | 0.179 | 0.838 | 0.810 |
| **SGU (ours)** | 2.6 | 0.523 | 2.5 | 0.194 | 0.860 | 0.866 |
| **TGU (ours)** | 2.5 | **0.631** | 2.4 | **0.169** | **0.857** | **0.871** |
| **Ground Truth** | 2.2 | - | 2.5 | - | - | - |

style of unseen speakers. For the NG and KL method, the gradient for the forget set became unbounded during training, causing the model to fail. The NG method performs poorly, showing high WER and low SIM scores on both sets. The KL method shows lower SIM score on forget set. However, significantly high WER suggests the model has learned to return inaudible noise for the forget set rather than a different voice. This is likely due to the entanglement between speaker style and linguistic content in the pre-training process, which makes it challenging for this method to disentangle the two aspects effectively.

Among all methods evaluated, TGU consistently achieves the best results, aligning strongly with our unlearning objectives. The SIM-F falls within the range of 0.169, which corresponds closely to the similarity scores observed between actual audio samples from different speakers. This demonstrates that TGU effectively generates voices distinct from the forget speaker prompts. While SGU also exhibits some level of success in reducing similarity for the forget set, it is significantly less effective than TGU, especially in maintaining performance on the remain set. Notably, TGU maintains an average SIM score of 0.631 for the remain set, showing only a 2.8% decrease compared to the original model, indicating a high level of retention for the original speaker identity's style. In contrast, SGU suffers a substantial drop of 21%, struggling to preserve the model's ability to replicate the remain speakers. For detailed information on the ground truth SIM values, refer to Appendix C.

In terms of WER, both TGU and SGU achieve results comparable to the original model, indicating that they do not compromise the correctness of speech generation. However, TGU outperforms SGU overall, proving to be the most effective unlearning method by balancing the dual goals of forgetting specific speaker identities while retaining the capability to generate high-quality speech for remain speakers.

*Table 2.* Quantitative results on LibriSpeech test-clean evaluation set (-R) and the forget evaluation set of (-F). $k$ refers to the number of forget speakers in the forget set. Please refer to Appendix E for the result of statistical significance analysis.

| Methods | WER-R ↓ | SIM-R ↑ | WER-F ↓ | SIM-F ↓ |
|---|---|---|---|---|
| **SGU** ($k$=1) | 2.7 | 0.586 | 2.8 | 0.173 |
| **SGU** ($k$=3) | **2.9** | 0.566 | 2.7 | 0.209 |
| **SGU** ($k$=10) | 2.6 | 0.523 | 2.5 | 0.194 |
| **TGU** ($k$=1) | **2.3** | 0.624 | **2.5** | 0.164 |
| **TGU** ($k$=3) | 2.9 | 0.626 | 2.3 | 0.159 |
| **TGU** ($k$=10) | 2.5 | 0.631 | 2.4 | 0.169 |
| **Ground Truth** | 2.2 | - | 2.5 | - |

**Randomness.** The rightmost two columns in Table 1 represent spk-ZRF results conducted on remain set and forget set. To grasp unlearned model's behavior, randomness on data with no knowledge of, the goal is to exhibit high spk-ZRF on forget set while performing similar to original model on the remain set. It should be recognized that a spk-ZRF too low on the remain set is undesirable as it means the model simply has learned to act in a consistent way, offering room for back traceable results.

Interpreting spk-ZRF alongside WER and SIM, we can notice that the behaviors of NG and KL fail to unlearn the speaker identity of forget set. While low SIM-F scores can be misleading, spk-ZRF successfully functions to depict that NG and KL show consistent generation for forget speakers. A spk-ZRF lower than the original model implies that the model fails to act in a way an unlearned model should. Rather, the model is simply responding with a same overfitted behavior - generating with no preservation of linguistic knowledge. This aligns with our analysis previously made; models unlearned with NG and KL fail to penalize only on the speaker identity, causing overall poor model performance.

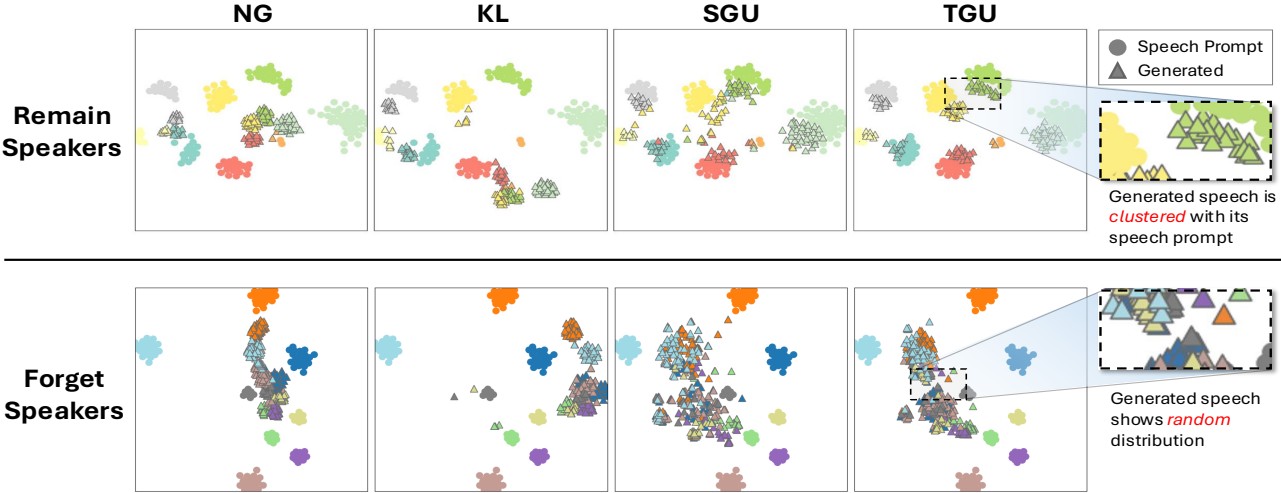

*Figure 3.* t-SNE analysis comparing NG, KL, SGU, and TGU for remain and forget sets. Samples from the same speaker are represented with the same color, where circles indicate actual speaker embeddings and triangles represent the embeddings of the model-generated speech. Ideal unlearned model should generate speech samples of remain speakers similar to its speech prompts; while generated speech samples of forget speakers should show random distribution - no correlation with any identity.

*Table 3.* Quantitative results on LibriSpeech test-clean evaluation set (-R) and the out-of-domain LibriTTS forget evaluation set of (-F).

| Methods | WER-R ↓ | SIM-R ↑ | WER-F ↓ | SIM-F ↓ |
|---|---|---|---|---|
| Original | 2.7 | 0.649 | 5.1 | 0.678 |
| SGU | 2.9 | 0.602 | 5.5 | **0.157** |
| TGU | **2.5** | **0.630** | **5.3** | 0.186 |
| Ground Truth | 2.2 | - | 5.9 | - |

Evaluated on randomness, SGU and TGU both show increased randomness across the forget set, while maintaining lower spk-ZRF across the random set. It can be acknowledged that both methods respond to the forget set with significant randomness in generation of speaker voices, while retaining knowledge across the remain set. TGU outperforms all other methods on spk-ZRF-F, exerting random speaker identities across the forget set.

**Scalability.** Table 2 shows that both SGU and TGU successfully unlearn the target speakers while preserving intelligibility on the remain set (-R). Notably, even when scaling from speakers of different sizes, both methods continue to yield solid results, with TGU displaying almost no performance degradation. In contrast, SGU suffers from a drop in similarity scores as more speakers are removed. On the scalability of guided unlearning approaches, this indicates that both methods can maintain similar levels of unlearning and speech quality regardless of the number of forget speakers.

**Out-of-Domain Unlearning.** In Table 3, we report evaluated results of our suggested unlearning methods under the scenario of preventing generation of a out-of-domain (OOD) speaker, where the speaker was not present in the pre-train dataset. Both SGU and TGU successfully unlearns speaker identities of forget speakers, with TGU maintaining average SIM-R of 0.630. Aligning with in-domain unlearning scenario, where the forget speaker was present in the pre-train dataset, SGU suffers a drop with 0602 and the highest WER for both remain (-R) and forget (-F). Both methods achieve results that indicate effective speaker identity unlearning even for speakers that were never seen during training.

### 5.3. Analysis

**Visualization.** Figure 3 illustrates the results of t-SNE, focusing on the model outputs for eight speakers selected from each set. The speaker embedding vectors of the input speech prompt and its resulting generated outputs were used for this analysis. For the forget set, SGU and TGU both showed that the embedding vectors of generated speech are intermixed, regardless of the prompt used. Both unlearning methods effectively remove the ZS-TTS system's ability to mimic forget speakers. In contrast, for the remain set, TGU demonstrated strong clustering among the embeddings of prompt and generated speech, showing consistent results for each speaker. SGU failed to achieve the same degree of clustering, with some embedding vectors intermixing rather than forming tight clusters. This indicates that TGU better preserves the performance of the original ZS-TTS system. NG and KL embeddings failed to cluster for remain speakers, and to show random distribution for forget speakers - suggesting poor unlearning performance overall.

*Table 4.* Human assessment on Librispeech test-clean evaluation set (-R) and the forget evaluation set (-F).

| Methods | CMOS-R ↑ | CMOS-F ↑ | SMOS-R ↑ | SMOS-F ↓ |
|---|---|---|---|---|
| Original | $0.00_{\pm 0.00}$ | $0.00_{\pm 0.00}$ | $4.47_{\pm 0.38}$ | $4.44_{\pm 0.36}$ |
| SGU (ours) | $-0.15_{\pm 0.27}$ | $-0.53_{\pm 0.28}$ | $3.12_{\pm 0.83}$ | $1.45_{\pm 0.31}$ |
| TGU (ours) | $\mathbf{-0.02}_{\pm 0.19}$ | $\mathbf{-0.45}_{\pm 0.23}$ | $\mathbf{4.67}_{\pm 0.26}$ | $\mathbf{1.28}_{\pm 0.24}$ |
| Ground Truth | $1.00_{\pm 0.26}$ | $0.22_{\pm 0.29}$ | $3.70_{\pm 0.70}$ | $3.89_{\pm 0.69}$ |

**Human Subjective Evaluation.** Table 4 presents the qualitative results for TGU and SGU. To compare the speech quality after applying speaker identity unlearning, we evaluated SGU and TGU using CMOS, with the original model as the baseline. The results show that TGU generates speech quality more similar to the original model compared to SGU, demonstrating its ability to better preserve high-quality speech generation. In terms of SMOS, TGU outperforms SGU on replicating voice styles for remain speakers. For forget samples, TGU produces voices that are more distinct from the prompt, effectively limiting the replication of the forget speakers. These results indicate that TGU effectively restricts the model's ability to mimic forget speakers and better preserves the performance of the ZS-TTS system. Please refer to Appendix G for detailed information on human evaluation including demographics, evaluation process and instructions.

## 6. Conclusion

In this paper, we applied and analyzed machine unlearning techniques for the first time in the context of speaker identity unlearning in Zero-Shot Text-to-Speech (ZS-TTS). Unlike traditional unlearning methods, randomness is incorporated to ensure that a model has forgotten its knowledge and ability to process the audio prompts of forget speakers. TGU effectively neutralizes the model's responses to forget speakers and limits the model's ability to replicate unwanted voices, while maintaining the performance of original ZS-TTS system. Our experiments showed that TGU results in only a 2.6% decrease in speaker similarity (SIM) for remain speakers, while maintaining competitive word error rate (WER) scores compared to the original model. Furthermore, we introduce a new metric to evaluate the lack of knowledge and trained behavior on the forget speakers, spk-ZRF. This metric evaluates randomness in voice generation to assess how effectively the unlearned model prevents reverse engineering attacks that could expose a speaker's identity.

## Impact Statement

Our work is dedicated to ensure safety in usage of Zero-Shot Text-to-Speech models by preventing the generation of voices belonging to individuals who do not consent to having their identities replicated. Although our work ad-

dresses the need for individuals to opt out of voice replication, determining how to handle similar voices raises complex questions. In striving to protect the privacy of a single individual, one could unintentionally restrict beneficial TTS capabilities to others whose voices resemble the forget set. Balancing personal privacy rights and broader technological benefits is at the heart of this tension. Also, techniques for ensuring speaker identity unlearning must be verifiable and transparent. Providing evidence that the model no longer replicates a forgotten identity requires both quantitative evaluation and subjective analysis. In light of the current situation—where many models remain closed due to concerns about misuse—we believe our work marks a new chapter in safeguarding individuals, paving the way for broader availability in the future. We aim to foster a deeper ethical discourse and encourage further research on responsibly handling ZS-TTS.

## Acknowledgements

This work was supported by the Institute of Information & Communications Technology Planning & Evaluation (IITP) under multiple grants funded by the Korea government (MSIT), including the ICT Creative Consilience Program (IITP-2025-RS-2020-II201821), the Information Technology Research Center (ITRC) support program (RS-2021-II212052), AI Graduate School Support Program (Sungkyunkwan University) (RS-2019-II190421), Artificial Intelligence Graduate School Program (Korea University) (RS-2019-II190079), the Artificial Intelligence Innovation Hub (RS-2021-II212068), AI Research Hub Project (RS-2024-00457882), AI Excellence Global Innovative Leader Education Program (RS-2022-00143911), and Development of Quantum Sensor Commercialization Technology (RS-2025-02217613).

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

# A. Experiment Settings

## A.1. Dataset Details

For the training set, we utilized the LibriHeavy dataset (Kang et al., 2024), which contains approximately 50,000 hours of speech from 7,000 speakers. To create the forget set, 10 speakers were randomly selected from the dataset. To avoid any bias in speaker selection, we first analyzed the distribution of audio duration per speaker in the LibriHeavy dataset. The lower and upper quartiles of audio duration per speaker were 440 seconds and 4,603 seconds, respectively. We randomly sampled 10 speakers whose audio durations fell within this range. For each selected speaker, approximately 300 seconds of audio was randomly chosen as the evaluation set, while the remaining audio was designated for the unlearning training set. The selected speakers are: *789*, *1166*, *3912*, *5983*, *6821*, *7199*, *8866*, *9437*, *9794*, and *10666*.

To evaluate the performance of the existing ZS-TTS model, specifically its ability to replicate the voices of unseen speakers, we used the LibriSpeech test-clean set (Panayotov et al., 2015). It is important to note that there is no overlap between the speakers in the LibriSpeech test-clean set and those in LibriHeavy (Kang et al., 2024). Following the experimental setup outlined in the original VoiceBox paper (Le et al., 2024; Wang et al., 2025), for both the forget and remain evaluation sets, a different sample from the same speaker was randomly selected, and a 3-second segment was cropped to be used as a prompt.

## A.2. Data Preprocessing

Speech is represented using an 80-dimensional log Mel spectrogram. The audio, sampled at 16 kHz, has its Mel spectral features extracted at 100 Hz. A 1024-point short-time Fourier transform (STFT) is applied with a 10 ms hop size and a 40 ms analysis window. A Hann windowing function is then used, followed by an 80-dimensional Mel filter with a cutoff frequency of 8 kHz. We used the Montreal Forced Aligner (MFA) (McAuliffe et al., 2017) to phonemize and force-align the transcripts, utilizing the MFA phone set, a modified version of the International Phonetic Alphabet (IPA), while also applying word position prefixes.

## A.3. Model Configurations

We applied both baseline machine unlearning methods and the proposed method to VoiceBox (Le et al., 2024), using the same configuration. The audio feature generator is based on a vanilla Transformer (Vaswani, 2017), enhanced with U-Net style residual connections, convolutional positional embeddings (Baevski et al., 2020), and AliBi positional encoding (Press et al., 2022). This model has 24 Transformer layers, 16 attention heads, and an embedding/feed-forward network (FFN) dimension of 1024/4096, with skip connections implemented in the U-Net style.

## A.4. Duration Predictor and Vocoder

We used the regression version of duration predictor proposed in (Le et al., 2024). The duration predictor has a similar model structure to the audio model, but with 8 Transformer layers, 8 attention heads, and 512/2048 embedding/FFN dimensions. It is trained for 600K steps. The Adam optimizer was employed with a peak learning rate of 1e-4, linearly warmed up over the first 5K steps and decayed afterward. HiFi-GAN (Kong et al., 2020), trained on the LibriHeavy (Kang et al., 2024) English speech dataset, is employed to convert the spectrogram into a time-domain waveform.

## A.5. Pre-training

Following (Le et al., 2024), we trained the original Voice model for 500K steps. Each mini-batch consisted of 75-second audio segments, and the Adam optimizer was employed with a peak learning rate of 1e-4, linearly warmed up over the first 5K steps and decayed afterward. All training was conducted using mixed precision with FP16.

## A.6. Inference Configurations

During inference, classifier-free guidance (CFG, (Ho & Salimans, 2022; Le et al., 2024)) was applied as follows:

$$\hat{v}_t(w, x, y; \theta) = (1 + \alpha) \cdot v_t(w, x_{ctx}, y; \theta) - \alpha \cdot v_t(w; \theta) \tag{13}$$

where $\alpha$ is fixed at 0.7, as specified in the original paper. Refer to Appendix F for information on the impact of $\alpha$.

We utilized the `torchdiffeq` package (Chen, 2018), which offers both fixed and adaptive step ODE solvers, using the default midpoint solver. The number of function evaluations (NFEs) was fixed at 32 for both the evaluation stage and the

generation of $\bar{x}$ in the proposed method. The Ground Truth for WER is obtained by transcribing the target speech using the Automatic Speech Recognition (ASR) model (Hsu et al., 2021), then comparing the ASR result to the target speech transcription.

## B. Unlearning Implementations

### B.1. Teacher-Guided Unlearning

The Teacher-Guided Unlearning (TGU) model was trained for 145K steps for 1 and 10K steps for 2. Each mini-batch included 75-second audio segments. The Adam optimizer was employed with a peak learning rate of 1e-4, which was linearly warmed up during the first 5 K steps and subsequently decayed throughout the remainder of the training. To facilitate the unlearning process, samples from the forget set $x^f$ were randomly selected with a 20% probability in each mini-batch.

### B.2. Sample-Guided Unlearning

To apply Sample-Guided Unlearning (SGU) in the ZS-TTS system, we set up the training process such that when a forget sample $x^f$ is provided, a random retain sample $x^r$ is selected as the target for training. To train VoiceBox, both speech data and aligned text segments are required. However, as discussed in Section 4.1, it is not naturally feasible to collect utterances from different speakers that share the same alignment. To address this, the SGU training was set up as follows: Let $y^f$ and $y^r$ represent the corresponding text segments for $x^f$ and $x^r$, respectively. We generated a mask corresponding to the length of $x^r$, training the model to predict $x^r$ based on this masked input. The text segments $y^f$ and $y^r$ were concatenated along the time axis and used as input, with the same process applied to the other input components, such as $w^f$ and $w^r$. During the training phase, the model was fine-tuned using 145K steps for 1 and 10K steps for 2. Additionally, forget samples $x^f$ and remain samples $x^r$ were selected and trained in a 2:8 ratio.

### B.3. Exact Unlearning & Fine-Tuning

The Exact Unlearning method was trained with the same configuration as the pre-training, except that only the dataset $D^r$ was used. Similarly, the Fine Tuning method involved additional training for 145K steps, exclusively using the dataset $D^r$.

### B.4. Negative Gradient

Implementation of Negative Gradient (NG) method follows that of (Thudi et al., 2022). On the pre-trained VoiceBox model, we provide only the samples from the forget speaker set $F$. The loss is inverted to counteract loss minimization previously occurred in the pre-trained model's weights. Given that approaches based on reversing the gradient often suffer from low model performance and unstable training, we searched for learning rate with best evaluation score {1e-5, 1e-6, 1e-7, 1e-8}. For evaluation, we use the checkpoint of 9.5K fine-tuned with Adam optimizer with a peak learning rate of 1e-8, linearly warmed up over first 5K steps and decayed after.

### B.5. Selective Kullback-Leibler Divergence

Numerous studies have adopted a loss function that focuses on utilizing a teacher-student framework with selective Kullback-Leibler divergence loss (Li et al., 2024; Chen & Yang, 2023). We implement this loss so the student model is fine-tuned to maximize KL-divergence between teacher and student output when $x^f$ is given as input, and minimize when $x^r$ is given :

$$L_{\text{KL}} = \lambda(\theta(x^r, y^r)\|\theta^-(x^r, y^r)) - (1-\lambda)(\theta(x^f, y^f)\|\theta^-(x^f, y^f)) \tag{14}$$

where $\lambda$ is a hyper-parameter between 0 and 1 to balance the trade-off. Similar to NG, unbounded reverted loss on KL-divergence is prone to low model performance. We searched for learning rate with best evaluation score from {1e-5, 1e-6, 1e-7, 1e-8}, and $\lambda$ from {0.5, 0.8}. For evaluation, we use the checkpoint of 32.5K fine-tuned with Adam optimizer with a peak learning rate of 1e-8, following warm up and decay of previous methods using $\lambda = 0.5$.

# C. Speaker Similarity in Real Samples

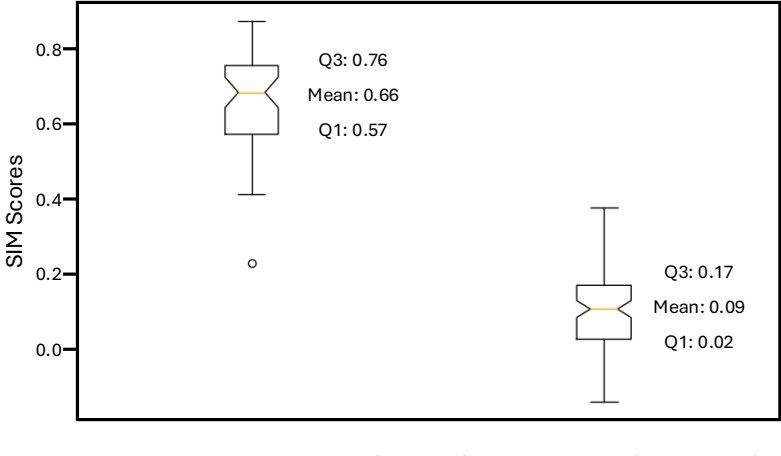

*Figure 4.* Boxplot of speaker similarity on same speaker's and different speakers' audio. Each are evaluated with 100 pairs of random speech audio in LibriSpeech test-clean subset.

From the LibriSpeech dataset, we make extensive analysis to get a grip of actual speaker similarity scores between pairs of audios from the same speaker, and that consisting of different speakers. For the SIM of same speakers, we retrieved random 100 pairs of audio, each pair comprised of different audio from random speaker. For the SIM of different speakers, similarly, we retrieved random 100 pairs of audio, with each pair comprised of audio from different speakers.

As shown in Figure 4, audios with same speaker's voice return SIM with 0.66 as mean, 0.57 and 0.76 each being lower and upper quartiles. With different speakers, mean of SIM is 0.09, lower and upper quartiles are 0.02 and 0.17. We take these values into consideration when evaluating Table 1 and Table 2. While actual values can have a wider range, we focus on the lower and upper quartiles as a primary boundary to achieve in unlearned models.

## D. Quantitative Results Over the Training Process

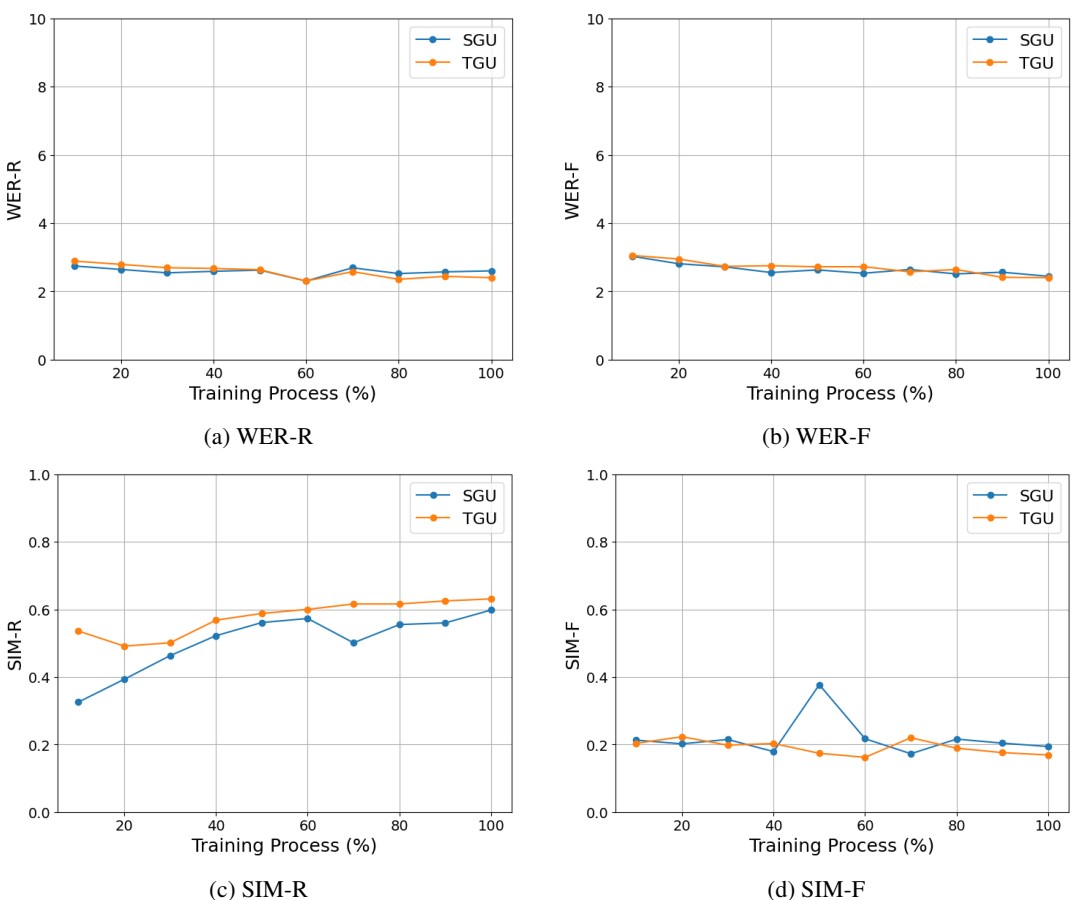

*Figure 5.* Quantitative results for SGU and TGU across different training stages. The top row shows the WER for both methods, while the bottom row displays the SIM results at each stage of the training process.

Figure 5 depicts the training process of our two proposed methods : SGU and TGU in Table 1. We evaluate the unlearning model's checkpoints at every 10% of full iterations. Notably, SIM score for the forget set declines quickly within first 10% of steps. However, SIM score for the remain set also declines in the early unlearning process - with the remaining process improving SIM-R.

Also, for WER scores for both remain set and the forget set remains relatively stable for both SGU and TGU. This suggests that guided unlearning method is highly effective in maintaining model performance in generating accurate speech on the given target text. It can also be interpreted that guided unlearning method is successful in disentangling speaker specific speech features from model's knowledge of correct speech generation.

## E. Statistical Significance of Experiments

Table 5 depicts the statistical significance analysis results of the paper's main experiments. The results reported in Table 1 and Table 2 were evaluated with a one-way ANOVA to assess how the unlearning method influences content correctness (WER), speaker similarity (SIM), and randomness in speaker identity (spk-ZRF). In Table 1, the analysis reveals a significant effect of the method on all metrics, demonstrating that the chosen unlearning strategy impacts content accuracy and speaker similarity. By contrast, in Table 2, low significance is observed in WER, indicating that two of our methods are comparable in terms of generating correct content. Nonetheless, the significant difference in SIM confirms that TGU is a more effective method for speaker identity unlearning.

*Table 5.* One-way ANOVA $F$ statistics for the effect of unlearning method. The *within-methods* degrees of freedom are $df_2 = 768$ for analyses on remain speakers (-R) and $df_2 = 1188$ for forget speakers (-F). ***$p < .001$.

| Tables | WER-R | SIM-R | WER-F | SIM-F | spk-ZRF-R | spk-ZRF-F |
|--------|-------|-------|-------|-------|-----------|-----------|
| Table 1 | 3900.01*** | 3275.76*** | 71.64*** | 501.71*** | 116.31*** | 807.97*** |
| Table 2 | 2.71 | 174.58*** | 2.80 | 7.44*** | - | - |

## F. Impact of $\alpha$

*Table 6.* Quantitative results based on the alpha value of CFG during the TGU inference process

|  | WER-R $\downarrow$ | SIM-R $\uparrow$ | WER-F $\downarrow$ | SIM-F $\downarrow$ |
|--|------|------|------|------|
| $\alpha = 0.0$ | 3.4 | 0.552 | 2.6 | 0.265 |
| $\alpha = 0.3$ | 2.6 | 0.583 | 2.3 | 0.198 |
| $\alpha = 0.7$ | **2.4** | **0.631** | 2.4 | **0.169** |
| $\alpha = 1.0$ | 2.5 | 0.629 | **2.4** | 0.187 |

In the CFG used during inference, $v_t(w; \theta)$ does not incorporate linguistic information $y$ or the surrounding audio context $x_{ctx}$, making it relevant to our formulation. To assess the impact of CFG on unlearning, we experimented with different values of $\alpha$. Table 6 presents the results of these experiments.

According to the results, when $\alpha$ is set to 0, removing the influence of $v_t(w; \theta)$, the model showed the highest SIM-F value, indicating increased reliance on $x_{ctx}$. On the other hand, when $\alpha$ was set to 0.3 or higher, the model consistently produced lower SIM-F values.

## G. Qualitative Evaluation Instruction

Table 7 and Table 8 present the instructions used for evaluating CMOS and SMOS in the qualitative assessment. Both the CMOS and SMOS evaluations were conducted with 25 participants.

*Table 7.* Comparative mean opinion score (CMOS) Instruction

---

**Introduction**
Your task is to evaluate how the quality of two speech recordings compares,
using the Comparative mean opinion score (CMOS) scale.

**Task Instructions**
In this task, you will hear two samples of speech recordings, one from each system.
The purpose of this test is to evaluate the difference in quality between the two files.
Specifically, you should assess the quality and intelligibility of each file in terms of
its overall sound quality and the amount of mumbling and unclear phrases in the recording.

**You should give a score according to the following scale:** -3 (System 2 is much worse)
-2 (System 2 is worse)
-1 (System 2 is slightly worse)
0 (No difference)
1 (System 2 is slightly better)
2 (System 2 is better)
3 (System 2 is much better)

---

*Table 8.* Similarity mean opinion score (SMOS) Instruction

---

**Introduction**
Your task is to evaluate how similar the two speech recordings sound in terms of
the speaker's voice.

**Task Instructions**
In this task you will hear two samples of speech recordings.
The purpose of this test is to evaluate the similarity of the speaker's voice between
the two files.
You should focus on the similarity of the speaker,
speaking style, acoustic conditions, background noise, etc.

**You should give a score according to the following scale:**
5 (Very Similar)
4 (Similar)
3 (Neutral)
2 (Not very similar)
1 (Not similar at all)

---

### G.1. Demographics of Human Evaluators

To assess the quality of synthesized speech, we conducted quantitative evaluation with total of 25 participants. Participants were recruited for individuals physically and cognitively capable of normal activities with ages between 20 and 45 years with high proficiency in English. Recruitment and study procedures adhered to Institutional Review Board guidelines, and all participants provided informed consent. Additionally, all participants were general listeners with no prior expertise in audio or speech synthesis.

### G.2. Evaluation Conditions

All participants completed a brief instructive session with an evaluator to familiarize themselves with the evaluation criteria. Evaluation was conducted in a quiet enclosed environment with the same listening device and volume levels, under the instructions of Table 7 and Table 8. Each evaluation took less than 10 minutes.

## H. Experiment on Unlearning Robustness

While Table 1 shows that TGU has effectively unlearned in overall, we go through extensive experiments to evaluate unlearning robustness. Figure 6 illustrates how TGU unlearned model behaves on remain speakers' speech prompts with various similarity scores to a forget speaker's speech prompt. As unlearning specifically on forget speakers is our objective in speaker identity unlearning, we expect the model to clearly classify forget speakers and remain speakers despite possible resemblances of each other.

For the x-axis, we identified speech prompts in remain set and the highest speaker similarity (SIM) score with any forget speech prompt. Then, the same remain speech prompts were used to generate speech with TGU unlearned model. The y-aixs was then obtained, by comparing the speech prompt with its TGU generated output speech. The results are visualized on 6.

A Pearson correlation analysis was conducted to assess the relationship between the similarity of remain speech prompts to forget speech prompts (x-axis) and the similarity of remain speech prompts to TGU-generated speech output (y-axis). The obtained statistic is 0.1396 while the p-value is 0.0003. This indicates a weak positive correlation with statistical significance, meaning that TGU generated speech is generally independent of the remain samples' similarity to forget speakers. Had the model not been robust and mistreated remain samples as forget speaker samples, there would have been a strong negative correlation.

In Table 9, we further assess the model's robustness by comparing its behavior on remain speakers with similar vocal characteristics to forget speakers. First, we compute the speaker similarity (SIM) between utterances from the two groups. We select utterances whose similarity to any forget-speaker utterance exceeds 0.40 and use these as prompts in our evaluation.

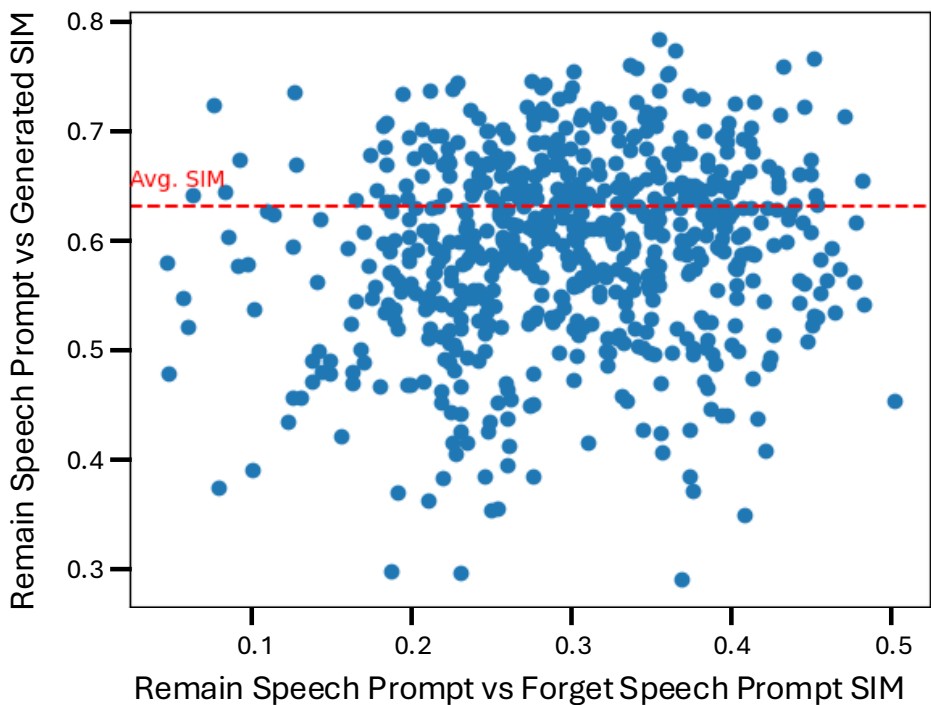

*Figure 6.* Robustness scatterplot of TGU on remain speakers. The x-axis represents the maximum SIM score between the remain speech prompt and forget speech prompt to depict the level of similarity between a remain speaker and a forget speaker. The y-axis represents the similarity score between the remain speech prompt and its resulting generated output using TGU. The red dashed line indicates average SIM score for all remain speech prompts in the evaluation set.

*Table 9.* Quantitative results on LibriSpeech test-clean evaluation set (-R) which show high speaker similarity to any forget speaker utterances (exceeding 0.40 in SIM).

| Methods | WER-R ↓ | SIM-R ↑ |
|---|---|---|
| **Original** | 4.96 | 0.637 |
| **NG** | 6.67 | 0.393 |
| **KL** | 8.78 | 0.316 |
| **SGU (ours)** | 5.70 | 0.411 |
| **TGU (ours)** | **4.70** | **0.622** |
| **Ground Truth** | 2.94 | - |

*Table 10.* Transient noise removal results on LibriSpeech test-clean set

| Methods | WER↓ | SIM↑ |
|---|---|---|
| **Clean speech** | 4.3 | 0.689 |
| **Noisy speech** | 47.9 | 0.213 |
| **Original** | **2.4** | **0.666** |
| **TGU (ours)** | 2.5 | 0.641 |

Even when a remain speaker's voice closely resembles that of a forget speaker, TGU maintains its performance with the original model. This demonstrates TGU's ability to preserve the identity of remain speakers while effectively neutralizing traces of forget speakers. In contrast, NG, KL and SGU significantly drop SIM-R (drops of 0.226–0.321), suggesting unlearning using these methods may trigger a trade-off that sacrifices remain speaker speech synthesis.

## I. Experiment on General Tasks

To provide deeper insights on how TGU unlearning may affect model performances on general tasks where ZS-TTS is used, we experiment the original model and TGU on transient noise removal.

### I.1. Transient Noise Removal

ZS-TTS can be applied in tasks where editing is required to remove undesired noise in speech datasets. To prevent having to go through repetitive and inefficient recording to obtain clean speech, ZS-TTS can generate clean audio for the noisy segment. We follow experimental settings of (Le et al., 2024) to analyze how TGU unlearned model performs on the task of transient noise removal.

From LibriSpeech test-clean dataset samples of durations 4 to 10 seconds, we construct noise at a -10dB signal-to-noise ratio over half of each sample's duration. Table 10 suggests that TGU provides comparable performances to that of the original model. While seemingly low, diminished model performances on transient noise removal is present relatively to the original model. We suggest that this is a trade-off from successful unlearning. While the model has unlearned to generate voice characteristics of the forget dataset, smaller knowledge-base and implemented randomness could have affected its reconstructing abilities.

### I.2. Diverse Speech Sampling

Being able to generate diverse speech is also an important feature of ZS-TTS models as it ensures realistic and high-quality speech that resembles natural distributions. This is necessary in applications such as speech synthesis or generating training data for speech related tasks (e.g., Automatic Speech Recognition). The diversity of generated speech samples is measured with Fréchet Speech Distance (FSD) as suggested in (Le et al., 2024). From generated speech samples, we extracted self-supervised features using 6th layer representation of wav2vec 2.0 (Baevski et al., 2020). The features were reduced to 128 dimensions with principle component analysis and used to calculate the similarity of distributions with real speech. High FSD indicates lower quality and minimal diversity, while low FSD refers to high quality and more diversity. For this experiment, $\alpha$ is set to 0 to ensure more diversity. Ground truth FSD is obtained by partitioning the LibriSpeech test-other set into half while ensuring equal distribution of data per speaker across both subsets

Experimental results in Table 11 show that FSD increases in TGU unlearned model. Because this task does not require input audio prompts, diverse speech sampling relies relatively heavier on datasets used to train the model. Implementing machine unlearning and thus inducing forgetting of specific speakers causes a trade-off in model's diversity. Meanwhile, it is noticeable that TGU achieves a lower WER in this case. We can infer that TGU obtains robustness in relatively noisy dataset comparable to the Original model.

## J. Recovery Experiment

Table 12 illustrates an experimental result on whether an unlearned model is recoverable to its original state. Aligning with our motivation to make ZS-TTS models safe, we presume a scenario of a privacy attacker who attempts to retrieve the

*Table 11.* Diverse speech sampling results on LibriSpeech test-other evaluation set

| Methods | WER ↓ | FSD ↓ |
|---|---|---|
| **Ground truth** | 4.5 | 164.4 |
| **Original** | 8.0 | **170.2** |
| **TGU (ours)** | **7.9** | 177.8 |

*Table 12.* Quantitative results for recovery experiments on unlearned models. WER and SIM evaluation follows the procedures of Table1.

| Methods | Recover Steps | Audio per Spk | WER-R ↓ | SIM-R ↑ | WER-F ↓ | SIM-F ↑ |
|---|---|---|---|---|---|---|
| Original | - | 15 min | 2.1 | 0.649 | 2.1 | 0.708 |
| TGU | - | 15 min | 2.5 | 0.631 | 2.4 | 0.169 |
| TGU | 36.25K | 15 min | 4.23 | 0.303 | 2.5 | 0.735 |
| TGU | 14.5K | 1 min | 4.61 | 0.226 | 2.8 | 0.162 |

original model parameters. We train the TGU unlearned checkpoints on all 10 of forget speaker's dataset to recover the original model. We also presume a practical scenario and attempt to recover the model performance using average of 1 minute for each speaker.

When given audio duration of 15 minutes for the forget speakers, the model fails to generalize over other speakers, hence, failing to mimic voices other than the forget speaker's. Additionally, the recovered model is more likely to generate wrong speech content as shown with higher WER in both remain set and the forget set. This process resembles fine-tuning a Text-to-Speech model for specific speakers rather than true recovery. Consequently, the original ZS-TTS model cannot be restored, and the attacker is essentially leveraging transfer learning to create a forget speaker-specific TTS model. However, with enough training data, the attacker could achieve similar results using any other non-zero-shot TTS model. We also consider a scenario where an attacker has access to only 1 minute of the forget speaker's voice sample. In this case, the model parameters also remain unrecoverable. The model also fails to generate forget speaker's voice. The model loses its zero-shot abilities hence the performance at early steps. Therefore, in practical scenarios where an attacker may attempt to train the model to clone an individual's voice with short sample of speech (e.g., voice phishing), it would not be feasible to recover the model or successfully generate the forget speaker's voice.

## K. Reproducibility Experiment

Table 13 illustrates the reproducibility of our experiment of Table 1 using a different dataset, LibriTTS (Zen et al., 2019). We pre-trained the VoiceBox model on LibriTTS for 500K steps and 10 speakers were randomly selected as forget set. When performing unlearning for only 10K steps (7% of pre-training steps), results of TGU illustrate effective unlearning while maintaining content accuracy.

## L. Inference Samples

Figures 7 and 8 show the Mel-spectrograms for the ground truth, original VoiceBox, SGU, and TGU inference results on forget speaker samples. These figures represent samples from speakers *789* and *6821*, respectively. The ground truth Mel-spectrogram corresponds to the audio where the same speaker as the prompt reads the same transcription.

*Table 13.* Quantitative results for reproducibility experiments using LibriTTS as pre-train dataset.

| Methods | Unlearn Steps | WER-R ↓ | SIM-R ↑ | WER-F ↓ | SIM-F ↑ |
|---|---|---|---|---|---|
| Original | - | 3.2 | 0.610 | 6.3 | 0.503 |
| TGU | 10K | 3.3 | 0.548 | 6.4 | 0.184 |

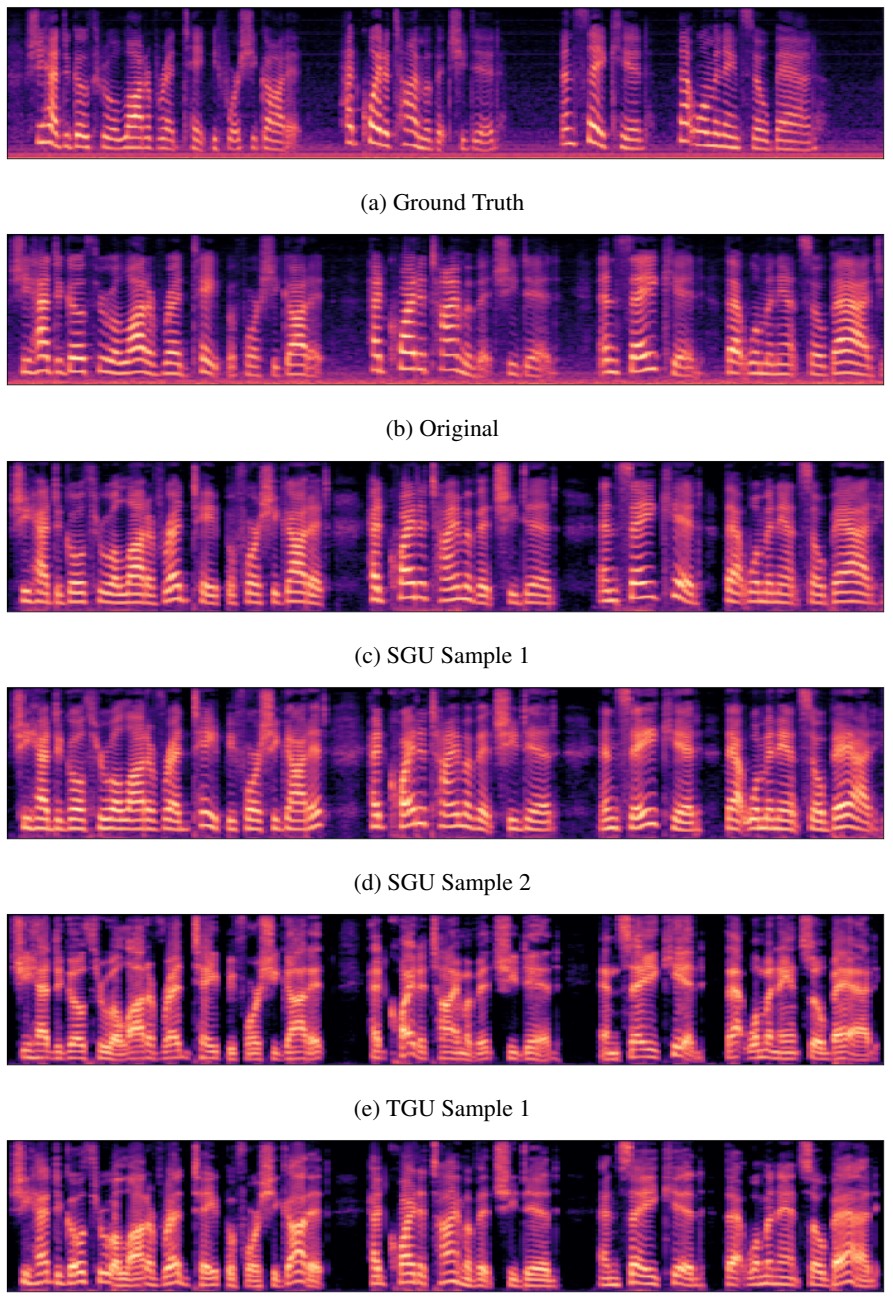

(a) Ground Truth

(b) Original

(c) SGU Sample 1

(d) SGU Sample 2

(e) TGU Sample 1

(f) TGU Sample 2

*Figure 7.* Mel-Spectrogram Comparisons: GT, Original, SGU Samples, and TGU Samples for the forget speaker *789*

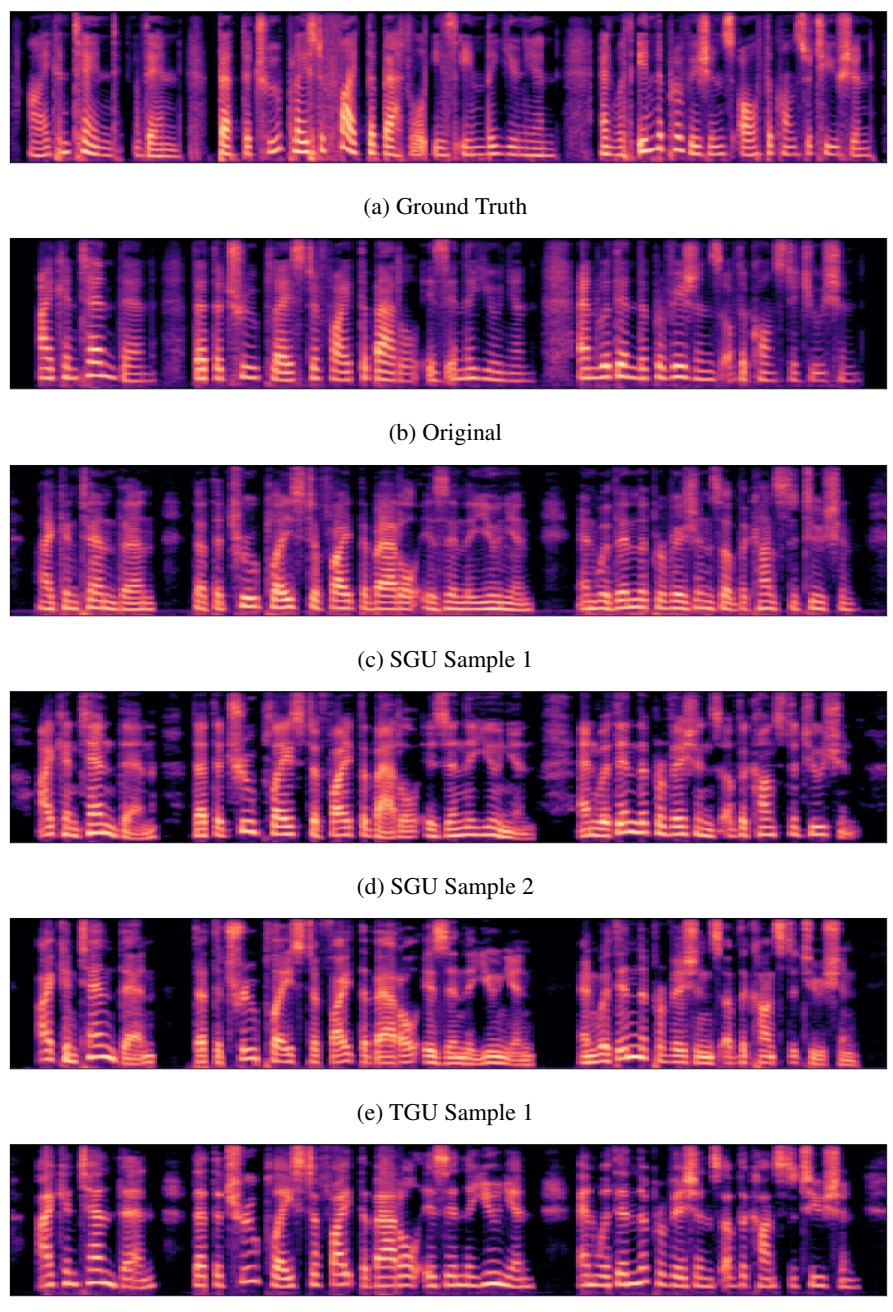

(a) Ground Truth

(b) Original

(c) SGU Sample 1

(d) SGU Sample 2

(e) TGU Sample 1

(f) TGU Sample 2

*Figure 8.* Mel-Spectrogram Comparisons: GT, Original, SGU Samples, and TGU Samples for the forget speaker *6821*

