# OpenReview forum: "Do Not Mimic My Voice : Speaker Identity Unlearning for Zero-Shot Text-to-Speech"
_ICML.cc/2025/Conference — ICML 2025 poster_

### Official Review · Reviewer_87b9 · 2025-02-28

**Overall Recommendation:** 3

**Summary:**

This paper proposes a method to exclude specific speakers from zero-shot text-to-speech to preserve voice privacy. It utilizes a guided learning approach in which some other speakers' voices are used in mask-based learning, especially a teacher-guided learning method in which the target guidance is generated from random noise. A new evaluation metric, spk-ZRF, is also proposed. The effectiveness of this method was confirmed in the evaluation using the LibliHeavy dataset.

**Claims And Evidence:**

The proposed teacher-guided learning and spk-ZRF measure is significantly novel and the evaluation experiments confirmed their effectiveness.

**Essential References Not Discussed:**

None.

**Experimental Designs Or Analyses:**

The details of human subject evaluation are unclear.

**Methods And Evaluation Criteria:**

1) How can the system know if the input audio prompt is from the speaker with the Forgot ID in the proposed unlearning framework?
     I am afraid it is difficult to identify speakers accurately from such small data.

2) The ideal method is Exact Unlearning, which requires a significant computational cost. Therefore, the proposed approach's computational costs should also be evaluated. The training and testing stages require extra computational costs.

**Other Comments Or Suggestions:**

None.

## update after rebuttal ##

Thank you for the rebuttal.
I did not change my score.

**Other Strengths And Weaknesses:**

1) The proposed methods' SIM-F is much lower than that of Exact Learning in Table 3. This result indicates that users may easily guess whose voices are excluded, which may also be a privacy problem. I believe it should be the same as Exact Learning.

2) The similarity measure may not be perfect for this task. Human listeners may use other cues to identify speakers, so a subjective test to determine a speaker should be needed.

**Questions For Authors:**

How many subjects are used,  and how much data are used, in the human subject evaluation?

**Relation To Broader Scientific Literature:**

This paper focuses on speaker identity in text-to-speech, but similar problems will happen in many other media.

**Theoretical Claims:**

The proposed method is theoretically sound.

---

> ### Author Rebuttal · Authors · 2025-04-01
>
> We sincerely thank the reviewer 87b9 for the thoughtful and constructive comments.
>
> ## **1. Human Subjective Evaluation**
>
> We deeply agree with the reviewer on that speaker similarity metric may not perfectly reflect human perception. Speech consists of pitch, prosody, and unique speaker characteristics.  To better capture human perception, we employed CMOS (Comparative Mean Opinion Score) and SMOS (Similarity Mean Opinion Score) in Table 3.
>
> CMOS was evaluated by asking participants to compare audio quality of outputs from the Original (Exactly Learned) model and outputs from SGU, TGU, and ground truth.
>
> SMOS was evaluated by presenting participants with two speech samples, and asking whether the two voices sound similar. For ground truth cases, we used two samples from the same speaker. For other cases, we used one real sample and one synthesized sample generated with same speaker’s prompt.
>
> Then, a well unlearned model should have very low SMOS-F and SIM-F, while keeping comparable scores to the Original model in SMOS-R and SIM-R.
>
> We prepared 160 audio samples in total (10 randomly selected samples x 4 systems x 4 conditions). 25 participants with proficiency in English were recruited and compensated for the experiment. The detailed information on human evaluation are described in **Appendix F.**
>
> ## **2. Forget ID Detection**
>
> As pointed out, the challenge of detecting if the input prompt belongs to a forget speaker under limited data is a non-trivial. On top of that, our framework does not consist of an independent speaker classifier or encoder.
>
> Instead of relying on explicit forget speaker classifier or detector, our model implicitly identify the forget speakers by by directly updating its parameters through our proposed training framework. Specifically, when given a forget speaker’s prompt, the model generates speech with a voice different from that of the forget speaker by treating it as noise and suppressing its identity. Thus, as the model generates random voices for forget speaker prompts, it becomes practically infeasible to reconstruct their original voices, ensuring stronger privacy protection.We verify this with multiple experiments. With proposed spk-ZRF, we can evaluate that TGU acts significantly randomly on forget speaker prompt.
>
> To further evaluate the robustness of our method against potential false positives, we conduct experiments using remain speakers who have high speaker similarity with the forget speakers. The results confirm that our model reliably distinguishes remain speakers without mistakenly treating them as forget speakers. For detailed experimental results, please refer to Section 7. Similar Voice Handling in our response to reviewer (bkoi).
>
> ## **3. Computational Costs**
>
> Contrary to traditional machine unlearning, Exact Unlearning is not ideal nor practical in Zero-Shot TTS. Zero-Shot TTS models can replicate voices they have never seen during training;  simply removing the forget speaker’s samples from training data does not suffice. Therefore, a comparable method to unlearn speaker identities does not exist.
>
> Regarding the training cost, the Exact Unlearning and Original models were trained for 500K steps. When unlearning 1 speaker at a time, SGU and TGU for were trained for 10K steps (2% of pretraining steps) in Table 2. When unlearning 10 speakers at a time with LibriTTS dataset, TGU was trained for 10K steps, refer to Section 4. Generalizability in our response to Reviewer d6q4.
> Fine Tuning, SGU, and TGU for Table 1. were trained for 145K steps - with TGU showing convergence around 58K steps in Appendix D. Negative Gradient and Kullback-Libeler divergence were halted at early stages due to exploding gradients.
>
> Nevertheless, our method is not confined to a specific fine-tuning method. Therefore, in practicality, computational cost can be minimized using diverse accelerating methods in the fine-tuning process.
>
> ## **4. SIM**
>
> SIM is calculated using the cosine similarity between speaker embeddings extracted from a speaker verification model. This evaluation is widely utilized in TTS research domain [1]-[6]. We use the same speaker verification model as [1]-[5] for evaluation. For SIM, the goal is to maintain a high performance in terms of ZS-TTS model evaluation.
>
> For unlearning task, indeed SIM is not sufficient alone. To mitigate this limitation, we propose spk-ZRF as a new evaluation metric to measure model’s randomness in generation. Objective is for the model to “act as if they have no knowledge of forget speaker identity”. Having a high spk-ZRF indicates that the model is behaving randomly. Vice versa, a spk-ZRF lower than original model means the model has learned to act in a consistent way (e.g., synthesizing in a specific voice). Thus, we do not mark an arrow for spk-ZRF-R because maintaining original performance is ideal.
>
> Once again, we greatly appreciate the reviewer’s insightful feedback, which helped us further clarify and strengthen our work.

---

### Official Review · Reviewer_d6q4 · 2025-03-13

**Overall Recommendation:** 3

**Summary:**

The paper proposes Teacher-Guided Unlearning (TGU), a novel framework for removing specific speaker identities from Zero-Shot Text-to-Speech (ZS-TTS) models to mitigate privacy and ethical concerns related to unauthorized voice cloning. The method leverages controlled randomness guided by a teacher model during fine-tuning, which obscures specific speaker identities while retaining the model's capability to synthesize speech for non-forgotten speakers. Additionally, the paper introduces Speaker-Zero Retrain Forgetting (spk-ZRF), a new metric that quantifies how effectively a model "forgets" speaker identities. Extensive evaluations are presented, comparing TGU against multiple baseline methods, demonstrating promising results in unlearning effectiveness and retention of model performance on non-forgotten speakers.

**Claims And Evidence:**

The main claims made by the authors are supported by a clear experimental setup and a thorough quantitative evaluation. Key claims include:

(1) **Effectiveness of TGU in speaker unlearning**: Experiments clearly demonstrate that TGU effectively reduces the similarity scores (SIM-F) between forgotten speaker prompts and synthesized speech, which validates the core unlearning claim.

(2) **Retention of performance for remaining speakers**: The evidence shows that the similarity (SIM-R) and speech quality (WER-R) scores for retained speakers remain competitive relative to baseline methods.

However, the claim about the robustness of the method when dealing with forgotten speakers similar in vocal characteristics to retained speakers is less convincing, as experiments reveal noticeable performance degradation under these circumstances.

**Essential References Not Discussed:**

Overall, essential related literature has been adequately referenced. However, considering the recent advancements in generative model unlearning across various modalities (e.g., diffusion models), additional comparative discussion with generative unlearning methods in other domains—such as recent image generation unlearning techniques—might provide broader insights into the generality of the introduced TGU framework and highlight potential cross-domain methodological inspirations.

**Experimental Designs Or Analyses:**

The experiments conducted are generally sound and appropriate:

- The use of well-established baseline methods for comparisons provides strong comparative validity.
- The analysis of the new spk-ZRF metric is meaningful and clearly shows advantages over existing metrics in evaluating speaker-specific unlearning.

However, some limitations persist in experimental analysis:

- While data augmentation is introduced to handle shorter audio prompts (1 min per speaker), the observed performance degradation in these scenarios limits practical deployment.
- Robustness analysis for retained speakers sharing vocal characteristics with forgotten speakers reveals limitations, suggesting room for further methodological improvements in addressing this scenario.
- Recoverability experiments showing how easily unlearned speaker identities can be restored were valuable but somewhat preliminary; further systematic analysis would provide deeper insights.

**Methods And Evaluation Criteria:**

**Strengths**
- The introduction of the spk-ZRF metric represents a significant methodological contribution, specifically designed for evaluating speaker-specific unlearning in ZS-TTS, addressing a gap in existing evaluation methods.
- The experimental evaluation strategy is generally robust, encompassing comparisons with well-established baseline methods (e.g., Exact Unlearning, Fine-Tuning, SGU, KL, NG).

**Weaknesses**
- The proposed method demands substantial computational resources (roughly 145K fine-tuning steps) and requires relatively long audio samples (approximately 5 minutes per speaker) to achieve reliable unlearning results, raising concerns regarding its real-world applicability.
- While authors present attempts at reducing resource requirements (such as data augmentation for shorter audio), these measures do not yet fully resolve concerns around scalability and computational practicality.
Evaluations conducted solely on a single proprietary architecture (VoiceBox) limit generalizability and broader applicability. Demonstrating effectiveness on multiple, especially open-source, architectures would significantly strengthen the validity of results.

**Other Comments Or Suggestions:**

Please refer to the previous sections.

**Other Strengths And Weaknesses:**

**Strengths**
- The paper introduces a genuinely original problem (speaker identity unlearning in ZS-TTS), clearly motivated by real-world ethical/privacy implications.
- The spk-ZRF metric is innovative and represents a methodological advance, explicitly tailored for the presented problem domain.
- The presentation is clear and structured, with comprehensive supplementary details that facilitate a deeper understanding of the proposed method.

**Weaknesses**
- Practical feasibility remains a significant concern, given computational overhead, extensive fine-tuning requirements, and the sensitivity of the approach to the length of the speaker audio samples.
- The limited evaluation conducted exclusively on the VoiceBox architecture restricts the scope of applicability; future work must include broader evaluations across diverse architectures.
- The robustness issues uncovered in scenarios involving highly similar retained and forgotten speakers require more systematic investigation and possibly methodological refinement.

**Questions For Authors:**

- Could you explicitly clarify whether you conducted subjective evaluations with proper ethical clearances (e.g., IRB approval)? This issue was raised previously and remains critical to the ethical soundness of the work.
- Considering the significant computational overhead, could you briefly describe how feasible this approach would be in an industrial deployment scenario where numerous speaker identities might need unlearning frequently?
- Could you clarify the rationale behind choosing only a 10-speaker "forget set" given the total dataset size (~50k hours)? What prevents you from testing larger-scale forgetting scenarios (e.g., 100+ speakers)?

**Relation To Broader Scientific Literature:**

The paper provides a novel perspective within a nascent area—speaker identity unlearning in ZS-TTS. While general unlearning methods (NG, KL divergence-based unlearning, Exact Unlearning, and Fine-Tuning methods) are well-explored, TGU's application of randomness is novel and justified within the context. Nevertheless, broader literature on generative model unlearning (e.g., image domain unlearning like Stable Diffusion unlearning methods cited) shows that generative models can often manage effective unlearning with far fewer resources. Thus, this paper's methods appear somewhat costly and practically limited by comparison.

**Theoretical Claims:**

The paper does not include formal theoretical proofs, thus no theoretical claims were assessed.

---

> ### Author Rebuttal · Authors · 2025-04-01
>
> We sincerely thank the reviewer d6q4 for the constructive feedback. Below we address each concern and clarify the raised points:
>
> ## **1. Similar Voice and Robustness**
>
> Please refer to Section 7 in our response to reviewer (bkoi)
>
> ## **2. Computational Resources**
>
> Please refer to Section 1 in our response to reviewer (bkoi)
>
> ## **3. Scalability and Length of Audio Samples:**
>
> We acknowledge concerns related to scalability, particularly regarding audio length requirements. From the perspective of service provider and user, we assume that the forget speakers voluntarily request their voices to be forgotten and are willing to provide audio samples specifically for the unlearning process.To alleviate the scalability concern under this assumption, we conducted experiments using data augmentation techniques leveraging the original model to generate additional training samples from as little as 1 minute of audio per forgotten speaker (as detailed in the "TGU w/ Augmentation" scenario). The results indicate substantial improvement in scalability, achieving a competitive SIM-R of 0.612 and SIM-F of 0.334 at only 48K training steps.
>
> | Method | Steps | Total Duration per Forget Speaker | WER-R | SIM-R | WER-F | SIM-F |
> | --- | --- | --- | --- | --- | --- | --- |
> | TGU | 145 K | 15 min | 2.5 | 0.631 | 2.4 | 0.169 |
> | TGU | 48 K | 15 min | 2.7 | 0.481 | 2.7 | 0.198 |
> | TGU w/ Augmentation | 48 K | 1 min | 2.6 | 0.612 | 2.6 | 0.334 |
>
> Although these results currently show room for further optimization, ongoing exploration into advanced data augmentation is expected to continue enhancing performance and practical viability, thus minimizing audio length constraints effectively.
>
> ## **4. Generalizability**
>
> **Model Selection**
> We selected VoiceBox due to its state-of-the-art (SoTA) performance at the time of conducting our research. We acknowledge the reviewer's concern regarding the generalizability and comprehensiveness of our evaluations. Given that MaskGCT [4] and F5TTS [3] were very recently introduced, we recognize their importance. F5TTS shares significant similarities with VoiceBox, being based on Flow-Matching with analogous model structures. We are currently preparing experiments using F5TTS to address this concern.
>
> **Reproducibility on Different Dataset**
>
> To address the reviewer's concern, we conducted additional experiments using another dataset; LibriTTS [10]. We trained the VoiceBox model on LibriTTS and 10 speakers were randomly selected as forget set, following experimental settings of Appendix A.
>
> | Method | WER-R | SIM-R | WER-F | SIM-F |
> | --- | --- | --- | --- | --- |
> | Original | 3.2 | 0.610 | 6.3 | 0.503 |
> | TGU | 3.3 | 0.548 | 6.4 | 0.184 |
>
> Due to time constraints, only TGU was evaluated at **10k steps (7% of pre-training steps)**.
> We plan to comprehensively evaluate additional baseline methods, including recent unlearning techniques, and add these results in the final camera-ready version of our manuscript.
>
> ## **5. Recoverability Analysis:**
>
> Thank you for finding our recoverability experiments valuable. We provide further analysis on recoverability experiments with additional metrics. Frechet Speech Distance (FSD) is used to measure Zero-Shot TTS’s ability to generate high quality speech with natural distributions. Low FSD indicates high quality and diversity in generated speech. Refer to Appendix H.2 for more information on FSD.
>
> | Method | Forget Audio Duration | WER-R | SIM-R | WER-F | SIM-F | FSD | spk-ZRF-R |
> | --- | --- | --- | --- | --- | --- | --- | --- |
> | Original | - | 2.1 | 0.649 | 2.1 | 0.708 | 170.2 | 0.857 |
> | TGU | - | 2.5 | 0.631 | 2.4 | 0.169 | 177.8 | 0.857 |
> | TGU-recover | 1min | 4.6 | 0.226 | 2.8 | 0.162 |  |  |
> | TGU-recover | 15min | 4.2 | 0.303 | 2.5 | 0.735 | 189.5 | 0.846 |
> | Ground Truth |  | 2.2 |  | 2.5 |  |  |  |
>
> According to results, recovering the original model by re-training is not possible.
>
> First, when recovering with audio duration of 1 min, the model fails to mimic any speaker. It even shows lower performance on the forget set in SIM-F.
>
> Second, when recovering with audio duration of 15 min, the model overfits to the forget speaker. Instead of recovering to Original, the  model is trained to be a speaker-specific TTS model.
>
> ## **6. Human Evaluation:**
>
> Please refer Appx F. and Section 1 in our response to reviewer (87b9).
>
> ## **7. Scalability - Forget Size**
>
> Please refer to our response to reviewer bkoi, specifically the section titled ‘**5. Scalability** ’
>
> Again, we sincerely thank the reviewers for their insightful feedback, which will substantially strengthen the paper’s final version.

---

### Official Review · Reviewer_bkoi · 2025-03-15

**Overall Recommendation:** 3

**Summary:**

The paper introduces Teacher-Guided Unlearning (TGU), a method to remove specific speaker identities from zero-shot text-to-speech (ZS-TTS) models to address privacy concerns. TGU guides the model to generate random voice styles for "forgotten" speakers while retaining synthesis quality for others. A new metric, speaker-Zero Retrain Forgetting (spk-ZRF), measures randomness in generated voices for forgotten speakers. Experiments on Voice-Box show TGU reduces similarity to forgotten speakers while maintaining performance on retained speakers.

**Claims And Evidence:**

`TGU requires only 5 minutes of audio per forgotten speaker (line 320 page 6)’: The method assumes access to the original training data and extensive retraining (around 25% of pre-training iterations), which is impractical for real-world deployment.

**Essential References Not Discussed:**

N/A

**Experimental Designs Or Analyses:**

1. The biggest issue with the experiment is that the validation is conducted on only a closed-source model, VoiceBox.

2. The paper lacks comparisons with some of the latest machine unlearning methods, such as [A-C].

[A]Low Compute Unlearning via Sparse Representations, arxiv, 2023

[B]Unified Gradient-Based Machine Unlearning with Remain Geometry Enhancement, neurips, 2024

[C]SalUn: Empowering Machine Unlearning via Gradient-based Weight Saliency in Both Image Classification and Generation,iclr,2024

**Methods And Evaluation Criteria:**

1. A major weakness of this work is the lack of a clear practical use case. Given that the proposed method requires 5 minutes of the original training audio from the forget speaker and incurs significant computational overhead, its practicality remains questionable.

2. The proposed spk-ZRF metric evaluates privacy risks solely through similarity and randomness measures but fails to validate practical resistance to adversarial attacks (e.g., reconstructing original voices via adversarial examples). Additionally, its reliance on pre-trained speaker verification models introduces bias risks—if these models are flawed or biased, the metric’s reliability is compromised.

**Other Comments Or Suggestions:**

The spk-ZRF-R metric in Table 1 is missing an arrow indicator.

**Other Strengths And Weaknesses:**

Strengths

1. This is the first work on speaker unlearning in ZS-TTS and also proposes a new metric, spk-ZRF, to evaluate the reconstructability of the target speaker, offering a novel perspective to the unlearning field.

2. The paper is well-structured, with clearly defined symbols and concepts, making it easy to follow.

Weaknesses

1. TGU requires access to the original training dataset and extensive retraining (compared to the pre-training stage), which conflicts with real-world scenarios where data may be unavailable due to privacy regulations.

2. Experiments are conducted only on Voice-Box, a closed-source model. Recent ZS-TTS models (e.g., YourTTS[D], CosyVoice[E], MaskGCT[F], F5TTS[G]) are not tested, raising concerns about the generalizability of the proposed method.

## update after rebuttal
All of my concerns have been addressed in the rebuttal, I decided to improve my score.

**Questions For Authors:**

1. How does SGU/TGU scale to more than 10 (e.g., 50) forgotten speakers while maintaining retain-set performance? In image classification tasks, unlearning methods are typically evaluated by increasing the count of the forget set from 1 to N to assess their scalability.

2. Can SGU/TGU handle cases where a retained speaker’s voice is accidentally similar to a forgotten one? Would this scenario affect the generation quality of retained speakers whose voices closely resemble those in the forget set?

3. Why was SGU/TGU not tested on open-source ZS-TTS models (e.g., YourTTS[D], CosyVoice[E], MaskGCT[F], F5TTS[G]) or more datasets? At the very least, evaluating the method on additional datasets and open-source models would help ensure the reproducibility of the proposed approach.

[A]Low Compute Unlearning via Sparse Representations, arxiv, 2023

[B]Unified Gradient-Based Machine Unlearning with Remain Geometry Enhancement, neurips, 2024

[C]SalUn: Empowering Machine Unlearning via Gradient-based Weight Saliency in Both Image Classification and Generation,iclr,2024

[D]YourTTS: Towards Zero-Shot Multi-Speaker TTS and Zero-Shot Voice Conversion for everyone, PMLR, 2022

[E]Cosyvoice: A scalable multilingual zero-shot text-to-speech synthesizer based on supervised semantic tokens, arxiv, 2024

[F]Maskgct: Zero-shot text-to-speech with masked generative codec transformer, ICLR, 2025

[G]F5-tts: A fairytaler that fakes fluent and faithful speech with flow matching, arxiv, 2024

**Relation To Broader Scientific Literature:**

The contributions of this paper is about the safefy of AI mdols. (Speaker Identity Unlearning of text-to-speech models)

**Theoretical Claims:**

There are no theorems or propositions in this paper; therefore, there are no proofs.

---

> ### Author Rebuttal · Authors · 2025-04-01
>
> We thank the reviewer bkoi for the constructive feedback and valuable insights. We address the primary concerns as follows:
>
> ## **1. Practical Use Case**
> The 145K steps used during unlearning processes were chosen based on the common fine-tuning practices when adapting Flow-Matching-based pre-trained models to the Zero-Shot TTS domain [11]. In this context, such a step count is not considered excessive. Furthermore, as presented in Appendix D, we provide performance curves throughout the unlearning process. Notably, we observe that after approximately 58K steps (about 11% of the pre-training steps), the model already achieves sufficient unlearning performance with SIM-R and SIM-F scores reaching 0.588 and 0.174, respectively. Importantly, we also confirm that WER-R and WER-F remain stable throughout the entire unlearning process.
>
> [11] Generative Pre-training for Speech with Flow Matching, ICLR, 2024
>
> ## **2. Adversarial Attacks**
>
> We agree that additional adversarial robustness evaluations would strengthen our method's validity. We plan to experiment this using adversarial attack methods in TTS to validate spk-ZRF’s resistance in the following week.
>
> ## **3. Latest Unlearning Methods**
>
> We provide results of latest machine unlearning methods with best scores of each methods in unlearning a single identity.
> |methods|WER-R|SIM-R|WER-F|SIM-F|
> |---|---|---|---|---|
> |SFRON [12]|36.60|0.186|68.61|0.091|
> |SalUn [13]|2.9|**0.637**|**2.3**|0.695|
> |SGU (Ours)|2.6|0.523|2.5|0.194|
> |TGU (Ours)|**2.5**|0.631|2.4|**0.169**|
>
> The poor results in [12] and [13] are expected as per the domain characteristics of ZSTTS. Recent architectures do not learn ‘speaker identity’ and ‘content generation’ separately. The features are deeply entangled within model parameters. Thus, removing the ability to mimic specific speaker identities without harming knowledge for content generation comes with an implicit but challenging task ; fine-grained feature disentanglement.
>
> Current saliency-based unlearning methods perform poorly. In [4], even though remain and forget set were both used to create separate fisher maps - unlearning in forget set still impacted performance for remain set. In [5], random labeling with unaligned content as label is used alongside fisher map from forget set. In this case, the model was updated minimally.
>
> We agree with your insights in adapting latest machine unlearning methods that can be adopted alongside Guided Unlearning. We plan to experiment saliency-based approach on SGU and TGU in the following week.
>
> [12] Unified Gradient-Based Machine Unlearning with Remain Geometry Enhancement
> [13] SalUn:Empowering Machine Unlearning via Gradient-based Weight Saliency in Both Image Classification and Generation
>
> ## **4. Speaker Verification Model**
> Please refer to Section 2 in our response to 4th Reviewer (87b9)
>
> ### 5. Scalability
> Here, we provide an updated version of Table 2. K refers to the of forget speakers in the forget set.
> |Methods|WER-R|SIM-R|WER-F|SIM-F|
> |---|---|---|---|---|
> |SGU (K=1)|2.7|0.586|2.8|0.173|
> |SGU (K=3)|2.9|0.566|2.7|0.209|
> |SGU (K=10)|2.6|0.523|2.5|0.194|
> |TGU (K=1)|2.3|0.624|2.5|0.164|
> |TGU (K=3)|2.9|0.626|2.3|0.159|
> |TGU (K=10)|2.5|0.631|2.4|0.169
>
> To analyze the impact of the number of forget speakers on model performance, we conducted  analysis by varying the forget speaker set size. As shown, our model maintains robust performance.
>
> Importantly, using 10 forget speakers is not a small-scale scenario compared to prior generative unlearning studies [10,12-13], where the forget set sizes are generally 1, and at most, 3. We plan to report our active experiments on 50 forget speakers in the camera-ready version.
>
> [12]  Forget-Me-Not: Learning to Forget in Text-to-Image Diffusion Models. arXiv, 2023.
>
> [13] Unlearning Concepts in Diffusion Model via Concept Domain Correction and Concept Preserving Gradient, AAAI, 2025
>
> ## **6. Similar Voice and Robustness**
>
> In Appendix G, we provide a figure illustrating TGU's performance on remain speakers with varying similarity to forget speakers. Pearson analysis (F=0.1396, p>0.0003) validates that even when a remain speaker may have vocal features similar to a forget speaker, this does not affect model's performance for remain speakers.
> We also provide comparisons of model performance for robustness. We obtain SIM between samples of remain speakers and forget speakers. Then, we identified samples of remain speakers that show high similarity ( > 0.4 ) to forget speakers. We use these samples as input prompt for evaluation.
> We will update the project page and supplementary to contain samples of similar speakers, and the generated output of each method.
>
> ||WER-R|SIM-R|
> |---|---|---|
> |Original|4.96|0.6365|
> |NG|6.67|0.393|
> |KL|8.78|0.316|
> |SGU|5.70|0.411|
> |TGU|4.70|0.622|
> |Ground Truth|2.94||
>
> ## **7. Generalizability**
> Please refer to Section 4 in our response to 3rd Reviewer (d6q4)z
> ## **8.spk-ZRF**
> Please refer to Section 4 in our response to 4th reviewer

---

### Official Review · Reviewer_k36c · 2025-03-16

**Overall Recommendation:** 3

**Summary:**

The paper presents two machine unlearning methods for zero-short TTS, with the main one called teacher-guided unlearning (TGU). This is an under-explored topic/task, which is of interest and importance. It further proposes a new evaluation metric for this task. A set of experiments were conducted to validate the performance of the proposed method.

## update after rebuttal
I appreciate that the authors conducted additional experiments to address my comments. I will maintain my score.

**Claims And Evidence:**

The paper presents a set of experiments, which clearly support the claims.

**Essential References Not Discussed:**

Sufficient references.

**Experimental Designs Or Analyses:**

In Table 1, the reason why the TGU row and the spk-ZRF-R column is highlighted is because it is the same as original, while the others are highlighted because they give either lowest or highest numbers. Then, it is unclear why the TGU row and SIM-R column is highlighted with bold font.

In Table 1, no statistical significant test results are provided for WER and SIM metrics. Same in Table 2.

What is the ground truth for WER? How to obtain them?

No results for exact unlearning and fine-tuning are provided in Table 3.

Why the scores of SIM-F are higher than those of SIM-R for both Exact Unlearning and Fine Tuning methods? Good to discuss.

**Methods And Evaluation Criteria:**

Experiments were conducted on LibriSpeech. The experiments are generally considered sufficient.

Several metrics are used, including WER, SIM, spk-ZRF, CMOS and SMOS.

"As a baseline, we applied four different approximate machine unlearning methods .... First, the Exact Unlearning method"
- This is incorrect as "the Exact Unlearning method" is not an approximate machine unlearning method.

Speaker similarity metric is not defined.

A set of demos were provided. The comparison with NG and KL is convincing. However, the authors didn't provide examples for exact unlearning and fine tuning.

**Other Comments Or Suggestions:**

"for retain speakers" -> "for retained speakers"

x is an audio prompt uttered by s ∈S
->
x^s is an audio prompt uttered by s ∈S

**Other Strengths And Weaknesses:**

Strengths:
This work explores a topic that is under-explored in the literature.
It proposes a metric for this task.
The results obtained are convincing.

Weaknesses
The experimental part could be enhanced, including the presentation and result discussion.

**Questions For Authors:**

The authors state that "unlearning in ZS-TTS presents unique challenges because the model can replicate speaker identities in a zero-shot manner, even without direct training on specific speaker data." Then, how unlearning is able to help when a specific speaker data is not used for training? Or, is this still a machine unlearning problem/task?

When introducing randomness in synthesizing speech, the proposed methods deliberately avoid the specific speaker, which seems no longer to be machine unlearning. Also, does this exactly reveal what specific identities are avoided?

**Relation To Broader Scientific Literature:**

Primarily for speaker identity unlearning for TTS.

**Theoretical Claims:**

The paper does not have theoretical claims. The proposed methods were experimental evaluated.

---

> ### Author Rebuttal · Authors · 2025-04-01
>
> We deeply appreciate the reviewer k36c's insightful comments and thorough review. Below, we address each comment clearly and concisely.
>
> ## **1. Clarification and Typos**
> As you correctly indicated, Exact Unlearning is an exact method rather than approximate. We will correct this with other typos and clearly state the distinction for revision. For supplementary, samples from Exact Unlearning and Fine Tuning methods will be included.
>
> ## **2. SIM-R and spk-ZRF-R**
> Please refer to Section 4. of our answer to 4th reviewer 87b9.
>
> ## **3. Statistical Significance**
> Here we provide statistical significance test results for Table 1 and Table 2. The result of ANOVA test on SIM and WER validates that the unlearning methods are effect SIM and WER.
> In Table 2, low significance in WER-R and WER-F validates that SGU and TGU both maintain comparable content accuracy. However, significance in SIM show that TGU is a more effective method.
>
> ||WER-R|SIM-R|WER-F|SIM-F|
> |---|---|---|---|---|
> |Table 1|F(4, 768) = 3900.01 , p < 0.0001|F(4, 768) = 3275.76 , p < 0.0001|F(4, 1188) = 71.64, p < 0.0001|F(4, 1188) = 501.71, p < 0.0001|
> |Table 2|F(1, 768) = 2.71, p > 0.01|F(4, 768) = 174.58, p < 0.0001|F(4, 1188) = 2.80, p > 0.01|F(4, 1188) = 7.44, p < 0.0001|
>
> ## **4. Ground Truth for WER**
>
> Word Error Rate (WER) is widely used in speech synthesis domain to analyze accuracy in content generation [1]-[5]. The Ground Truth for WER in **Table 1** is obtained by transcribing the target speech using the Automatic Speech Recognition model [6], then comparing the ASR result to the  target speech transcription. The target speech is obtained from the test dataset LibriSpeech test-clean [7].
>
> [1] Voicebox: Text-Guided Multilingual Universal Speech Generation at Scale, NeurIPS, 2023
> [2] Neural Codec Language Models are Zero-Shot Text to Speech Synthesizer, Arxiv, 2023
> [3] F5-tts: A fairytaler that fakes fluent and faithful speech with flow matching, aAxiv, 2024
> [4] Maskgct: Zero-shot text-to-speech with masked generative codec transformer, ICLR, 2025
> [5] CosyVoice: A Scalable Multilingual Zero-shot Text-to-speech Synthesizer based on Supervised Semantic Tokens, 2024
> [6] HuBERT: Self-Supervised Speech Representation Learning by Masked Prediction of Hidden Units, 2021
> [7] Librispeech: An ASR corpus based on public domain audio books, ICASSP, 2015
>
> ## **5. Missing Results in Table 3**
> Thank you for your feedback. We initially focused on methods with noticeable changes in SIM-F, but we acknowledge the value of assessing Exact Unlearning and Fine Tuning.
> Following Institutional Review Board (IRB) guidelines to ensure ethical evaluation takes time, and we will add these results in camera-ready version.
>
> ## **6. Discussion on Higher SIM-F**
> While SIM-R measures the zero-shot voice replicability on completely unseen speakers [9], SIM-F is evaluated with the test set from data [8] the model was pre-trained on. In the case of Original, SIM-F is inevitably higher due to similarity in data distribution with the train set. Exact Unlearning and Fine Tuning both fail to unlearn due to model's zero-shot capabilities, thus for similar reasons, SIM-F scores are higher.
>
> [8] Libriheavy: a 50,000 hours asr corpus with punctuation casing and context, ICASSP, 2024
>
> ## **7.Unlearning Unseen Speakers**
> We experiment this Out-of-Domain Unlearning [9] scenario where the challenge precisely lies in the fact that ZS-TTS can replicate voices without explicit training data for speakers. We unlearn speaker identity never used in training and results demonstrate that our unlearning method is effective, even for unseen speakers.
>
> |Dataset|WER-R|SIM-R|WER-F|SIM-F|
> |---|---|---|---|---|
> |Original|2.1|0.649|5.13|0.678|
> |SGU|2.9|0.602|5.5|0.157|
> |TGU|2.5|0.63|5.32|0.186|
> |Ground Truth| 2.2||5.93||
>
> [9] Generative Unlearning for Any Identity. CVPR, 2024.
> [10] LibriTTS: A Corpus Derived from LibriSpeech for Text-to-Speech, Interppeech, 2019
>
> ## **8. Revisiting Machine Unlearning**
>
> We adopt the task of machine unlearning for Zero-Shot TTS, and introduce newly formulated problem as Speaker Identity Unlearning (SIU) in Section 3.2
>
> SIU directly adjusts model parameters to remove the capability to reconstruct forget speaker’s voice. This is consistent with recent trends in generative unlearning. A recent study [9] demonstrates that unlearning can be applied to prevent identities from being visually generated. Similarly, our work prevents identities from being synthesized upon request. Also, the goal of machine unlearning is to safeproof model parameters from revealing information. Same goal is applied in SIU, with notable results in preventing traceability of forget set. Our methods are consistent with machine unlearning's problem formulation; prevent revealing unwanted identities, while retaining model performance on remain set.

---

### Decision · Program_Chairs · 2025-05-01

**Decision:**

Accept (poster)

**Comment:**

This work proposes a Teacher-Guided Unlearning (TGU) which is a novel framework for removing specific speaker identities from Zero-Shot Text-to-Speech (ZS-TTS) models. It leverages controlled randomness via a teacher model to unlearn specific speaker identities while retaining the ability to emulate non-forgotten speakers. The work also proposes a new evaluation metric which generates random voice styles for "forgotten" speakers termed speaker-Zero Retrain Forgetting (spk-ZRF) thus measuring randomness in generated voices for forgotten speaker.

The reviewers consistently recognize this as the first work on speaker unlearning and the novelty of the proposed new metric, spk-ZRF. The work represents a significant methodological contribution and relies on robust evaluation strategy comprising comparisons with well-established baseline methods. The reviewers also expressed the following concerns:

- the unlearning method has a significant computation cost and requires considerable about of user data (approx 5 mins per speaker) which might limit it's real-world applicability.
- validation is conducted using closed-source model, VoiceBox which may further limit adoption.

Despite the mentioned concerns, the reviewers agree that this is a significant contribution to the community. I request the authors take into account the detailed feedback from the reviewers to improve the final draft of their submission.